# DETECTING DATA CONTAMINATION IN LLMS VIA IN-CONTEXT LEARNING

**Michał Zawalski**
NVIDIA
mzawalski@nvidia.com

**Meriem Boubdir**
NVIDIA
mboubdir@nvidia.com

**Klaudia Bałazy**
NVIDIA*

**Besmira Nushi**
NVIDIA
bnushi@nvidia.com

**Pablo Ribalta**
NVIDIA
pribalta@nvidia.com

## ABSTRACT

We present **Contamination Detection via Context (CoDeC)**, a practical and accurate method to detect and quantify training data contamination in large language models. CoDeC distinguishes between data memorized during training and data outside the training distribution by measuring how in-context learning affects model performance. We find that in-context examples typically boost confidence for unseen datasets but may reduce it when the dataset was part of training, due to disrupted memorization patterns. Experiments show that CoDeC produces interpretable contamination scores that clearly separate seen and unseen datasets, and reveals strong evidence of memorization in open-weight models with undisclosed training corpora. The method is simple, automated, and both model- and dataset-agnostic, making it easy to integrate with benchmark evaluations.

## 1 INTRODUCTION

Detecting contamination is essential for the integrity of LLM evaluation (Dodge et al., 2021; Maini et al., 2024; Carlini et al., 2021; Singh et al., 2024). Existing approaches, such as loss-based criteria (Shi et al., 2024; Zhang et al., 2024; Carlini et al., 2021), score calibration using external models (Maini et al., 2024; Carlini et al., 2021), and explicit overlap checks (Gao et al., 2021; Biderman et al., 2023), often require access to training data, extensive parameter tuning, and may fail to produce reliable, interpretable results for large models. This calls for an automated, scalable method applicable across diverse datasets and architectures.

To fill this gap, we introduce Contamination Detection via Context (CoDeC), a simple yet effective dataset-level method that detects data contamination via in-context learning. CoDeC builds on the observation that context sampled from a dataset seen during training provides no new information to the model, while unseen data can improve predictions as in few-shot learning (see Figure 1). Hence, the percentage of samples in a dataset for which adding context from the same dataset reduces confidence quantifies contamination. CoDeC requires only gray-box access to token probabilities, works with any dataset, and needs no prior knowledge of the model's training corpus.

Unlike traditional methods focused solely on strict membership inference, CoDeC detects contamination arising also from training on augmented or related data, such as synthetically generated shadow data, which exposes the model to distribution-specific cues. By capturing both direct memorization and contamination via related distributions, CoDeC proves valuable both for fair evaluation and guiding model development.

Through extensive experiments, we show that CoDeC clearly separates training from unseen datasets, achieving near-perfect dataset-level AUC 99.9% across many models, while baseline methods fail.

---

*Work done while at NVIDIA.

0The code to run CoDeC can be found at https://github.com/NVIDIA-NeMo/Evaluator/blob/main/packages/nemo-evaluator/examples/notebooks/contamination_detection_demo.ipynb

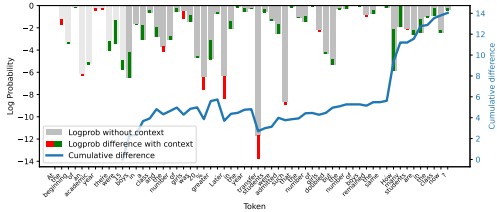 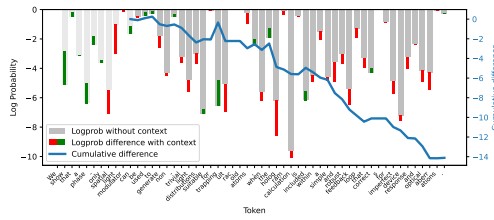

(a) Changes in model confidence on an unseen dataset.     (b) Changes in model confidence on a training dataset.

Figure 1: Change in model log-probabilities with and without context for the Pythia 1.4B model, evaluated on a sample from an unseen dataset (gsm8k, 1a) and a training dataset (ArXiv, 1b). Differences in confidence for consecutive tokens are consistent. See Appendix C.2 for details of this experiment.

We apply CoDeC to recent open-source models, evaluating on widely used benchmarks, finding multiple instances of potential contamination. Our ablations further demonstrate that CoDeC is stable across diverse datasets and model types, and can be further enhanced by scaling the context. CoDeC enables the community to better trust reported LLM results and supports the development of more reliable model evaluation practices.

In summary, our main contributions are the following:

- We introduce CoDeC, a simple model-agnostic approach that effectively detects data contamination in LLMs using in-context learning.
- We confirm the strong advantage of CoDeC over baselines through experiments on diverse models and datasets, analyze its properties, and propose practical guidelines for applying it.
- We use CoDeC to evaluate a variety of recent models on widely used benchmarks, revealing several instances of potential contamination.

## 2   CONTAMINATION DETECTION VIA CONTEXT

### 2.1   PROBLEM DEFINITION

Given a language model $M$ and a candidate dataset $\mathcal{D} = \{x_i\}_{i=1}^{N}$, where each $x_i$ is a text sequence, our goal is to quantify contamination, i.e. whether that dataset or similar data was in the training set of $M$ and the model is relying on memorization rather than generalization.

### 2.2   KEY IDEA

LLMs respond differently to in-context examples depending on prior exposure. When given an unseen dataset, adding in-context samples taken from that dataset usually improves the model's confidence, as it can generalize better with more information (Brown et al., 2020). However, if the model has memorized the dataset, the in-context samples not only provide little new information but also disrupt memorization patterns, leading to reduced confidence (Razeghi et al., 2022).

Thus, by comparing confidence levels with and without in-context learning across sample sequences, we can leverage these shifts to detect contamination. Note that no generation is involved, as we measure only logits of the given sequences to estimate confidence. We provide experimental evidence for those observations in Section 3 and Appendix C.2, and extended discussion in Appendix A.

### 2.3   CODEC PIPELINE

Our method (presented in Figure 2) consists of the following steps:

1. **Baseline prediction:** For each datapoint $x$ in the suspect dataset $\mathcal{D}$, obtain the model's average log-likelihood on the consecutive tokens of $x$.
2. **In-context prediction:** Sample $n$ additional examples $x_1, ..., x_n$ from $\mathcal{D} \setminus \{x\}$, prepend them to $x$ (creating a sequence $x_1|...|x_n|x$), and obtain the model's predictions on $x$ in this new context.
3. **Score computation:** Compute the difference in confidence $\Delta(x) = \text{logprob}_{\text{in-context}}(x) - \text{logprob}_{\text{baseline}}(x)$.

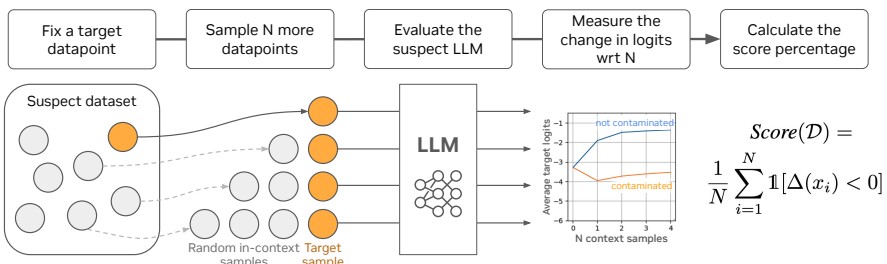

Figure 2: Overview of Contamination Detection via Context (CoDeC). For each dataset element, CoDeC augments the context with a few other samples from the same dataset. A decrease in the model's logits for the target sample indicates potential contamination. The overall contamination level is estimated as the fraction of samples exhibiting this effect.

4. **Aggregation:** Repeat the above for all $x \in \mathcal{D}$. The contamination score for the dataset is then

$$S_{\text{CoDeC}}(\mathcal{D}) = \frac{1}{N} \sum_{i=1}^{N} \mathbb{1}[\Delta(x_i) < 0]$$

where $\mathbb{1}$ is the indicator function.

## 2.4 PROPERTIES

CoDeC offers several properties that make it broadly applicable and easy to use. It produces intuitive, percentage-based contamination scores that are immediately interpretable and require no model- or dataset-specific calibration, avoiding the threshold tuning often needed in membership inference methods due to the arbitrary scales of their raw scores (Maini et al., 2024; Carlini et al., 2021; Shi et al., 2024).

The method works with any dataset that can be represented as a collection of text sequences, including standard training corpora and widely used benchmarks. It is model-agnostic and requires only gray-box access to token probabilities. CoDeC is also computationally efficient, needing just two forward passes per sample. A detailed discussion of its theoretical and empirical properties can be found in Appendix A.3.

## 2.5 WHY DOES CODEC WORK?

The effectiveness of CoDeC stems from several intuitive principles:

1. **Dataset-specific priors reveal contamination.** Models trained on a dataset internalize its unique style, structure, vocabulary, and implicit assumptions. If these priors are already memorized, additional in-context examples add little useful information.
2. **Context disrupts memorization.** For contaminated datasets, adding memorized in-context examples can interfere with memorized token sequences. The presence of patterns triggering memorization in the context confuses the model, leading to reduced confidence.
3. **ICL simulates finetuning dynamics.** Contaminated models resemble finetuned models near saturation – additional training yields minimal gains, whereas non-contaminated models adapt faster. CoDeC effectively measures how much learning capacity remains for the dataset. Figure 8 illustrates these dynamics.
4. **Interventions expose loss landscape.** Contamination is often linked to overfitting (Maini et al., 2024). Overfitted models sit in narrow local minima that are more easily destabilized by new context, while unseen data corresponds to flatter, more stable loss surfaces.

In the next section, we provide empirical evidence grounding those intuitions. Further discussion can be found in Appendix A.5.

## 3 EXPERIMENTS

In this section, we demonstrate that CoDeC consistently distinguishes between seen and unseen data, is stable across evaluation settings, and yields interpretable scores for model auditing.

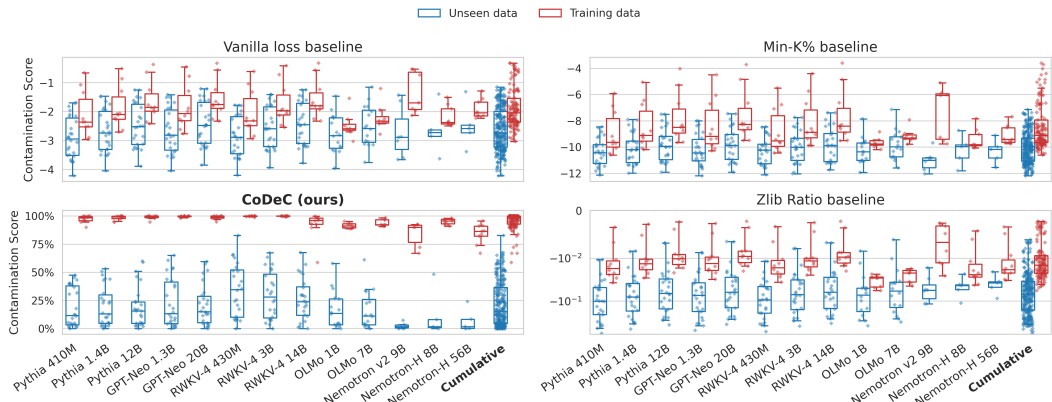

Figure 3: **CoDeC vs. baselines**; Contamination scores for training (red) and unseen (blue) datasets. Each point is a model–dataset pair. CoDeC achieves the best separation, enabling consistent classification across models and datasets.

**Models.** We validate CoDeC on a diverse suite of LLMs with publicly available weights and training data, enabling reproducibility and ground-truth verification. Our evaluation includes models trained on different corpora and spanning varied architectures: Pythia (Biderman et al., 2023), GPT-Neo (Black et al., 2021), and RWKV-4 (Peng et al., 2023), all trained on the Pile (Gao et al., 2021); OLMo (Groeneveld et al., 2024), trained on Dolma (Soldaini et al., 2024); Nemotron-v2 (NVIDIA et al., 2025a) and Nemotron-H (NVIDIA et al., 2025b), trained on Nemotron-CC (Su et al., 2024). Those models cover transformers, RNNs, and hybrid mamba-transformer architectures.

**Datasets.** For each model, we create a test bed consisting of (1) data known to be in its training set (e.g., subsets of the Pile) and (2) unseen data published after the model's training cutoff. The unseen data includes recent benchmarks, news, websites, etc. A complete list of the datasets can be found in Appendix B.

**Baselines.** We compare CoDeC against three common contamination detection methods: **Vanilla Loss** (Fu et al., 2024), scoring based on model loss; **Min-K%** (Shi et al., 2024), focusing on the most informative tokens; and **Zlib Ratio** (Carlini et al., 2021), which normalizes perplexity by sample entropy. More details can be found in Appendix C.1.

| Model | CoDeC (ours) | Vanilla loss | Min-K% | Zlib |
|---|---|---|---|---|
| Pythia 410M | **100.0%** | 75.0% | 76.2% | 92.3% |
| Pythia 1.4B | **100.0%** | 77.3% | 79.2% | 91.5% |
| Pythia 12B | **100.0%** | 76.9% | 82.3% | 92.3% |
| GPT-Neo 1.3B | **100.0%** | 77.3% | 79.2% | 90.8% |
| GPT-Neo 20B | **100.0%** | 76.9% | 83.5% | 92.7% |
| RWKV-4 430M | **100.0%** | 75.4% | 75.4% | 92.3% |
| RWKV-4 3B | **100.0%** | 76.5% | 79.6% | 91.9% |
| RWKV-4 14B | 99.6% | 77.3% | 81.5% | 92.7% |
| OLMo 1B | **100.0%** | 64.8% | 71.9% | 80.5% |
| OLMo 7B | **100.0%** | 65.6% | 72.7% | 78.1% |
| Nemotron v2 9B | **100.0%** | 83.7% | 97.6% | 98.2% |
| Nemotron-H 8B | **100.0%** | 80.0% | 73.3% | 88.0% |
| Nemotron-H 56B | **100.0%** | 82.2% | 86.7% | 92.0% |
| **Cumulative** | **99.9%** | 75.7% | 78.5% | 89.6% |

Table 1: AUC scores for separating seen vs. unseen datasets (Figure 3), computed per dataset[1].

## 3.1 MAIN VALIDATION

CoDeC cleanly separates seen from unseen data (see Figure 3), achieving **dataset-level AUC of 99.9%** across all evaluated models (see Table 1). In contrast, baselines fail to provide a reliable separation. Their scores for seen and unseen data overlap significantly, making them ineffective for confident contamination detection. The consistent and sharp separation achieved by CoDeC enables meaningful, model-agnostic comparisons.

To further validate CoDeC, we examined how scores for each dataset vary across different models. Figure 5 shows the distribution of CoDeC scores on various datasets for all models trained on the Pile corpus (extended results can be found in Appendix C.3). For most datasets scores are remarkably consistent, clustering around the dataset-specific mean located between 0 and 60%. Such stability suggests that major outliers in the distribution are indicators of at least partial contamination.

---

[1]While AUC, a parameter-free metric commonly used for evaluating MIA methods (Maini et al., 2024), is usually computed over individual samples, here it is computed over dataset-level scores. Details are provided in Appendix B.4.

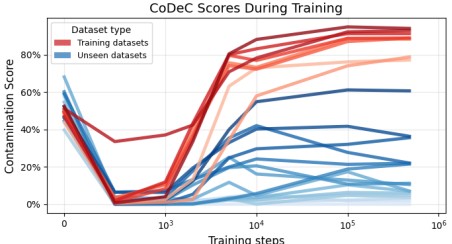

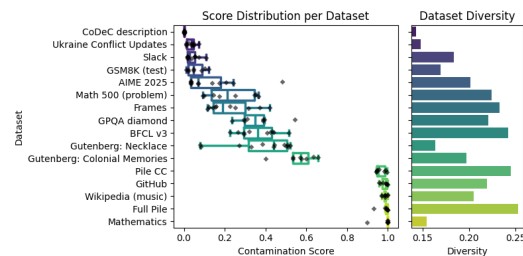

Figure 4: Changes in CoDeC scores over training of OLMo 7B. A list of the datasets evaluated in this experiment can be found in Appendix C.4.

Figure 5: Distribution of CoDeC scores per dataset, computed for models trained on the Pile. The last 5 datasets are parts of the training data. The full evaluation can be found in Appendix C.3.

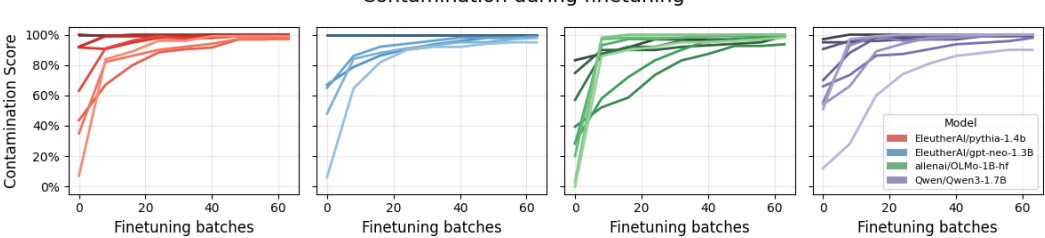

Figure 6: CoDeC scores during finetuning different models on 10 datasets (see Appendix B.3 for the list). Though the Qwen3 model does not disclose its training data, by finetuning we can confirm the effectiveness of CoDeC also for that architecture.

We also analyzed whether dataset diversity affects CoDeC scores by measuring the variance of text embeddings (see Figure 5). While a certain correlation exists, it is insufficient to explain the observed patterns, indicating that CoDeC captures deeper signals beyond simple diversity. Importantly, this finding suggests that scores even near $50\%$ should not be treated as contamination hints for highly diverse datasets, especially if the scores are consistent across models.

## 3.2 CONTAMINATION DURING TRAINING

Figure 5 tracks how CoDeC scores change during training of the OLMo 7B model, for which training checkpoints are publicly available. At initialization, scores are close to $50\%$ for all datasets, reflecting random delta-confidence fluctuations in an untrained model. By 1k steps, the model has acquired basic language understanding but has seen too little to memorize anything. Only the Stack-v4 dataset (code snippets and byte-like samples) drops only to $33\%$ at 1k steps because it does not align with general language skills, leaving memorization as the only viable strategy.

The key shift occurs between 1k and 10k steps, which makes just about $2\%$ of the full training. Contamination scores for training datasets rise sharply, while scores for unseen datasets settle near their final values. Beyond this point, up to the final 477k training steps, scores remain largely stable – indicating that the model has already seen enough data to recognize and memorize the distributions of its training sets.

Since CoDeC detects contamination in the very early stages of training, it is highly effective for identifying and preventing benchmark leaks, supporting more reliable model development. The complete list of datasets presented in Figure 4 can be found in Appendix C.4.

## 3.3 CONTAMINATING VIA FINETUNING

**Finetuned models approach** $100\%$ **CoDeC score on all datasets.** To further validate CoDeC, we performed a controlled finetuning experiment on four example models (Pythia, GPT-Neo, OLMo, and Qwen3) as shown in Figure 6. For each run, we took a model and finetuned it on a chosen dataset, both from its original training data and from unseen sources. In each case, the CoDeC

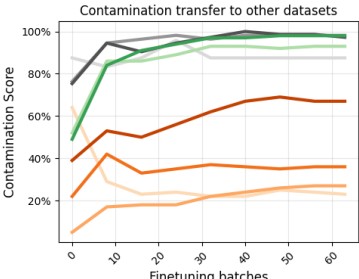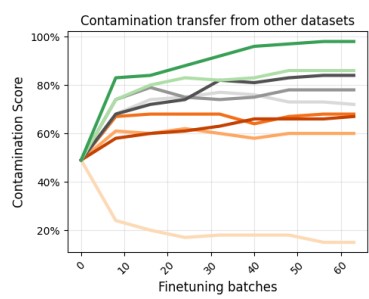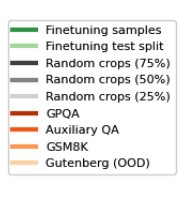

Figure 7: Contamination transfer between MMLU and related distributions, computed for the Pythia model. Evaluation on other distributions, including data mixes and noise injection can be found in Appendix C.5.

score consistently rose above $90\%$, showing a clear and stable increase after exposure to the data. Finetuning on a dataset clearly introduces contamination, and CoDeC reliably detects this effect.

This setup is independent of any uncertainty about the model's original training corpus. As a result, the evaluation can be applied to any model, such as Qwen3 (Yang et al., 2025), to confirm the applicability of CoDeC for recent models, even with undisclosed training data.

**Contamination transfer.**   Next, we explore how contamination from one dataset influences related datasets, and vice versa. Specifically, we analyzed the effect of finetuning a model on MMLU and measuring CoDeC scores on datasets with varying levels of similarity to MMLU, and conversely, training on related datasets and evaluating contamination on MMLU.

Figure 7 (left) shows how training on MMLU influences CoDeC scores on other datasets. Finetuning makes the model highly contaminated even with unseen questions from MMLU. Contamination scores remain high even when we randomly crop the evaluation data down to as little as $25\%$ of the original content (which was already short).

Contamination transfers modestly across QA benchmarks because MMLU shares question format, style, and structure with them. Exploiting these patterns can boost benchmark performance without reflecting broader knowledge. Accordingly, models contaminated with one QA dataset show slightly elevated CoDeC scores on related QA datasets, though far from the high-contamination range. No such transfer is observed for unrelated datasets (e.g., MMLU and Gutenberg books).

The right chart in Figure 7 shows the reverse: training on related datasets and evaluating on MMLU. Unsurprisingly, the relative impact is almost identical as in the first case. Even finetuning only on parts of the samples or otherwise augmented text drove the CoDeC scores high on the original MMLU. Importantly, this demonstrates that CoDeC cannot be cheated by simple data augmentations.

More evaluations for the contamination transfer experiment, including other augmentations (synthetic rephrases, added noise, whitespace changes, etc.), can be found in Appendix C.5.

**In-context learning reveals overfitting.**   We compared finetuning to adding in-context samples and found a close relationship between them (see Figure 8). Confidence curves obtained by finetuning with a moderate learning rate and by adding extra context follow an identical pattern, suggesting a similar influence on model behavior. Contaminated datasets are highly sensitive to the learning rate: even small updates cause early drops in confidence of the model. Unseen datasets are robust, with confidence improving even under very high learning rates. This pattern is consistent across models and datasets.

These results support the view that contamination is closely related to overfitting (Maini et al., 2024): memorized samples sit in high-variability, narrow minima where even small updates can destabilize predictions, whereas unseen samples lie in flatter regions that benefit from additional learning signals. CoDeC captures this distinction, which explains why the method is effective.

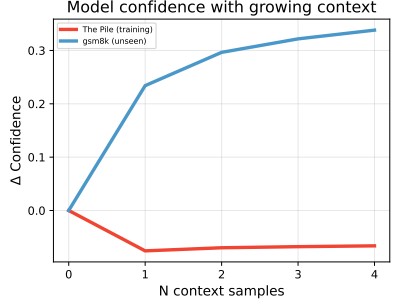
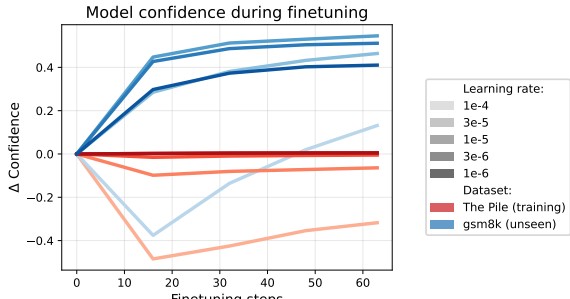

(a) Changes in model confidence with adding more samples to the context.

(b) Changes in model confidence during finetuning with various learning rates.

Figure 8: Relation between finetuning and in-context learning for contamination detection. Both plots show how the average probability of target tokens evolve as more learning signal is provided. For learning rate around $3e^{-5}$, the curves nearly overlap.

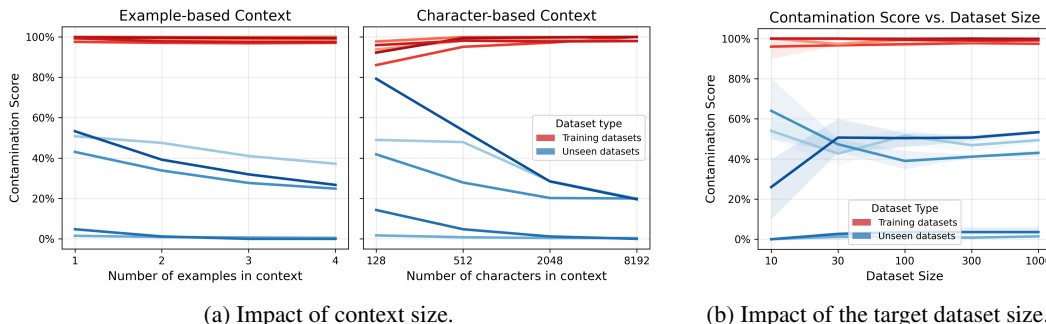

(a) Impact of context size.

(b) Impact of the target dataset size.

Figure 9: Ablation studies of CoDeC on the Pythia 1.4B model using 5 training and 5 unseen datasets (see Appendix B.3 for the list). Shaded regions show the range between the minimum and maximum scores across 5 runs.

## 3.4 ABLATIONS

**Context Size.** While our experiments use the simplest form of CoDeC with a single in-context sample, the method can be easily extended to include more context. Larger contexts yield clearer separation between seen and unseen data but also increase computational cost (see Figure 9a). In practice, a single in-context sample already provides strong signal while keeping inference efficient. However, adding more samples can further enhance performance if resources permit.

**Target Dataset Size.** A key practical factor is how many examples are needed for reliable contamination scores. CoDeC proves highly sample-efficient: as few as 100 examples already give stable estimates with low variance, making the method feasible even on small benchmarks. With 1,000 examples, the variance with respect to the sampled contexts falls below 1% (see Figure 9b). Since the method is statistical, each sample can be viewed as a Bernoulli trial to derive confidence bounds easily.

## 3.5 INTERPRETING CODEC SCORES

CoDeC captures how strongly model's predictions depend on memorized patterns and whether outputs are driven by internal distributional priors. Note that this can also occur if the model has been trained on identically distributed or closely related data (e.g., synthetically generated). Hence, while high scores do not prove that a dataset was in the training corpus, the method is informative for separating generalization capabilities of the model from specialization. Among models with similar benchmark accuracies, models with lower CoDeC scores should generalize better.

For highly contaminated datasets, scores typically cluster near $100\%$. While most unseen datasets are scored below $20\%$, some can reach as high as $60\%$. This may occur for two main reasons: (i) the dataset is highly related to the training data, resulting in partial contamination; or (ii) the dataset consists of extremely diverse samples, so in-context examples act like random noise. Hence, to separate dataset effects from true memorization, scores should be compared across multiple models on the same dataset.

**Best practices.** For robust contamination assessment, we suggest combining absolute thresholds with reference-based comparisons. High absolute scores (>80%) should be treated as contamination red flags, but significant deviations from other models' CoDeC scores should be equally alarming. For a broader discussion, see Appendix A.5.

### 3.6 BROADER APPLICATION OF CODEC

Experiments on models with known training data, presented in Section 3, confirm the reliability of CoDeC. With confidence in its validity, we subsequently applied CoDeC to a broad suite of over 40 recent models and analyzed their contamination scores across widely used benchmarks. The full evaluation can be found in Appendix D.

Most models fall well below the high-contamination area, suggesting they were not directly trained on benchmark data. Nevertheless, certain models score close to this threshold or remain clear outliers on specific benchmarks, which may indicate partial data leakage or substantial training on closely related synthetic or derivative content. Variants within the same model family tend to produce similar scores, reinforcing CoDeC's robustness across model types. The largest models generally achieve very low contamination scores, supporting the view that increased capacity facilitates generalization over memorization.

These findings underscore the support of CoDeC in identifying contamination risk, interpreting benchmark accuracy with appropriate caution, and informing fair model comparison.

## 4 LIMITATIONS AND FUTURE WORK

While CoDeC demonstrates robust and promising results, there are several areas for improvement. Scores may be affected by adversarial datasets, such as those with heavy duplication or mixtures of unrelated sources, although these issues do not compromise evaluations on standard benchmarks (see the discussion in Appendix A.3.2). In principle, it may be possible to engineer training strategies that bypass increases in CoDeC estimates. However, if such approaches preserve a model's ability to learn and prevent it from internalizing dataset-specific priors, they would represent a notable advance in their own right. Improving the criterion so that untrained models or unrelated data mixtures consistently yield scores well below $50\%$ could further enhance interpretability. Another direction is adapting CoDeC for strict membership inference at the sample level, enabling more fine-grained analysis. Finally, additional research is needed to fully explore the applicability of CoDeC across diverse model architectures and training regimes or proving theoretical guarantees for CoDeC scores.

## 5 RELATED WORKS

**Contamination detection.** Detection of training data contamination in LLMs is closely related to research on membership inference attacks (MIAs), which aim to determine whether a specific data sample was used in model training (Shokri et al., 2017). Early methods relied on training shadow models (Zhang et al., 2021; Yuan et al., 2023), which is prohibitively costly for modern LLMs. A simple approach to detecting contamination is based on thresholding loss (Yeom et al., 2018) or perplexity (Carlini et al., 2021). (Shi et al., 2024; Zhang et al., 2024) improve over those methods by focusing on the most informative tokens. Mattern et al. (2023); Mitchell et al. (2023) detect contamination by perturbing samples to check for memorization. Unlike loss-based approaches, CoDeC does not require model-specific threshold tuning and offer much better separability, which makes it particularly useful in practice.

Many approaches rely on reference models for calibrating the entropy (Min et al., 2024; Maini et al., 2024). Similarly, Carlini et al. (2021) proposes estimating entropy using zlib library. More recent

approaches leverage held-out data to train a model-calibrated contamination classifier (Maini et al., 2024; Bar-Shalom et al., 2025). CoDeC in a way follows the general structure of reference-based criteria, but it uses the same model as a reference, adapting to its internal confidence distribution.

Numerous approaches analyzed benchmark contamination in LLMs. For example, Yang et al. (2023) show that simple n-gram overlap de-duplication is insufficient, as paraphrases, translations, and variations of test samples can still leak into training data. Golchin & Surdeanu (2024) observed that some models, when given only the name of a benchmark, can recreate its questions. Oren et al. (2023) compare the loss when samples are presented in a canonical order versus a random order, under the assumption that a contaminated model will favor the former. Other approaches measure whether the model can reconstruct masked answers or even answer without being given the question (Deng et al., 2024; Balepur et al., 2024). However, such methods often suffer from low recall or rely on narrow assumptions about the targets, whereas CoDeC avoids these limitations entirely.

Recent studies prove that the problem of membership inference is inherently hard, especially in the era of LLMs trained on massive corpora and usually for just a single epoch, rendering most prior methods ineffective (Maini et al., 2024; Duan et al., 2024; Carlini et al., 2021). Instead, Maini et al. (2024) proposed dataset-level approaches to achieve meaningful results, but they require access to held-out identically distributed samples. CoDeC follows the dataset-level setup, though it works without additional constraints.

Beyond attacks, several defenses have been proposed to mitigate contamination detection. Jia et al. (2019) introduced adversarial filtering to reduce memorization. Shejwalkar & Houmansadr (2020) explored selective forgetting using differential privacy and gradient masking, while Mireshghallah et al. (2021) investigated privacy-preserving fine-tuning to obfuscate membership signals.

**In-context learning.** LLMs are known to improve when prompted few-shot (Brown et al., 2020). The connection between in-context learning and training is a widely studied relationship (von Oswald et al., 2023; Mosbach et al., 2023; Dai et al., 2023). CoDeC builds upon this work, and extends the understanding of how in context examples impact generation confidence differently when the model has already seen the text vs. not.

## 6 CONCLUSIONS

We introduced Contamination Detection via Context (CoDeC), a simple yet very effective method for detecting and quantifying training data contamination in large language models. By measuring how in-context examples from the same dataset affect model predictions, CoDeC distinguishes between datasets the model has memorized and those it has not. Our experiments show that CoDeC produces clear, interpretable contamination scores on a wide range of models and datasets, exposing evidence of memorization and overfitting even in open-weight models with undisclosed training data.

Unlike traditional membership inference methods, CoDeC requires no external references, dataset-specific tuning, or retraining, making it practical for large-scale, real-world evaluations. Its percentage-based scores are easy to interpret and integrate into benchmark reporting.

While high CoDeC scores ($> 80\%$) indicate likely contamination, moderate values should be compared across models to separate dataset effects from memorization. Importantly, the scores do not indicate only strict training membership but capture also the influence from rephrased, augmented, or otherwise highly related data, making CoDeC a practical tool for ensuring fair evaluation.

## 7 REPRODUCIBILITY STATEMENT

In the supplementary material, we include a notebook that enables running CoDeC on any chosen model and dataset. All models and almost all datasets used in our experiments are publicly available on HuggingFace, making our work fully reproducible. We strongly encourage readers to use the attached code to evaluate contamination on data of their choice.

## 8 LLM USAGE

We used LLMs to help us refine the wording of the manuscript, to examine related works, and as coding assistants during experiment development. However, these uses were limited in scope and did not play a significant role in the research itself.

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

## A DETAILED EXPLANATION OF CONTAMINATION DETECTION VIA CONTEXT

### A.1 KEY IDEA

Consider the following illustrative example. If a mathematician trains for the International Mathematical Olympiad (IMO) competition by solving all available IMO problems, their knowledge becomes highly contaminated with IMO-specific problem characteristics. Beyond the genuine mathematical knowledge acquired, exposure to the IMO problem distribution introduces several specific priors, such as: *all assumptions mentioned in the problem are necessary*, *the hypothesis requested to prove always holds*, *the problems are challenging yet formulated without advanced academic concepts*, etc. These properties are unique to IMO problems but not to general mathematical questions and texts. Therefore, leveraging such priors results in an unfair measure of one's general mathematical skills.

Now, suppose this mathematician is given free access to IMO problems during the competition. If they have already learned to solve those problems, such context is no longer useful for further learning. However, if the mathematician has learned from other sources and has never seen any IMO problem, they would benefit greatly from understanding the specifics of these problems, their structure, and implicit assumptions.

Following this example, Contamination Detection via Context (CoDeC) measures the contamination score by evaluating how access to the target dataset affects the model's predictions. For each datapoint in the dataset, CoDeC samples a few in-context examples from the same dataset and compare the average logits with and without this context.

If the model has not used the target dataset for training, it should benefit from the additional context, as these examples contain valuable information about the data distribution. Even if not directly related, they may share similarities in structure, style, vocabulary, topic, or other implicit common priors. Conversely, if the model was trained on that dataset, it already knows these priors, making the provided context less beneficial. Furthermore, if the model is overly confident due to memorized patterns, such as specific word sequences or frequent occurrences, introducing additional context (likely also memorized) should disrupt these patterns and negatively affect the confidence.

See the analysis in Appendix C.2 for empirical evidence of those observations.

### A.2 CODEC PIPELINE

Following the key idea described above, the complete pipeline of CoDeC consist of 4 simple steps:

1. **Baseline prediction:** For each datapoint $x$ in the suspect dataset $\mathcal{D}$, obtain the model's average log-likelihood on the consecutive tokens of $x$.

2. **In-context prediction:** Sample $n$ additional examples $x_1, ..., x_n$ from $\mathcal{D} \setminus \{x\}$, prepend them to $x$ (creating a sequence $x_1|...|x_n|x$), and obtain the model's predictions on $x$ in this new context. Focus solely on the probability of tokens in $x$, ignoring the context examples. While any number of context examples can be used, in our experiments we used one.

3. **Score computation:** Compute the difference in confidence $\Delta(x) = \text{logprob}_{\text{in-context}}(x) - \text{logprob}_{\text{baseline}}(x)$. Since the context is sampled randomly, the value of $\Delta(x)$ is subject to some variance. To achieve higher statistical significance, we average the $\Delta(x)$ values over 5 seeds, though even a single seed provides meaningful scores. Additionally, when computing the average log-likelihood, We ignore first 10 tokens to avoid the influence of switching between texts.

4. **Aggregation:** Repeat the above for all $x \in \mathcal{D}$. The contamination score for the dataset is then

$$S_{\text{CoDeC}}(\mathcal{D}) = \frac{1}{N} \sum_{i=1}^{N} \mathbb{1}[\Delta(x_i) < 0]$$

   where $\mathbb{1}$ is the indicator function.

In summary, to compute CoDeC scores, we compare the model's confidence with and without additional context for each sample. The final score represents the percentage of samples for which the influence of additional context is negative.

### A.3 Properties of CoDeC

#### A.3.1 Theoretical properties

CoDeC by its design has several desirable properties that make it useful in practice:

**Score as a percentage.**  CoDeC returns a contamination score as a percentage of dataset elements indicating contamination, making it intuitively understandable. Unlike classical approaches that compute scores in open ranges requiring references and scale knowledge, percentage points are easily grasped without in-depth analysis. Even the intuitive notion of *probability of contamination*, while not fully grounded, can be useful here.

**Works with any dataset.**  CoDeC can be computed for any set of strings, making it broadly applicable. In our evaluations, we used various data sources: standard training datasets (e.g., Wikipedia, Common Crawl), code (e.g., GitHub), QA benchmarks (e.g., MMLU, GPQA), math benchmarks (e.g., AIME, MATH-500), PDFs (e.g., ArXiv), books (e.g., Project Gutenberg), websites (e.g., global news, Amazon Reviews), files (e.g., Linux syslog), and more. While CoDeC is designed primarily for datasets, it is straightforward to transform a single text source into a dataset by splitting it into parts.

**Works with any gray-box model.**  Any model that outputs logits for a given text (including most models available on HuggingFace) can be used to compute CoDeC. The method is independent of the model's training specifics and architecture. It can even be applied to fully black-box models by empirically estimating target token probabilities, though this increases score variance unless we assume multiple repetitive calls and model stochasticity.

**Parameter-free.**  CoDeC does not require model-specific or dataset-specific parameters. Unlike classical MIA methods, which need a tuned classification threshold and deep model knowledge or external ground-truth data, CoDeC can be applied off-the-shelf to any model and dataset.

**Computationally efficient.**  To compute CoDeC scores, the model runs twice with at most twice longer samples, introducing minimal computational overhead. Using as few as 1000 samples already yields precise scores (see Figure 9b), hence it can be applied even to large corpora. Additionally, in-context learning is much faster than methods requiring finetuning or shadow model training.

#### A.3.2 Empirical properties

Furthermore, during our experiments we observed the following empirical properties of CoDeC:

**Robustness to text cropping.**  CoDeC scores remain stable when using partial training prompts. This property proves particularly valuable when evaluating single, long text sources (books, articles) that should be split into dataset components.

**Sensitivity to formatting variations.**  Artificial formatting changes (added labels, modified whitespace) reduce contamination scores even for training data. Understanding that sensitivity is crucial for reliable QA benchmark evaluation, where labels like *Question:*, *Answer:*, or instruction prompts are common. Incorrect labels (those not used during training) decrease scores, as expected: in-context samples provide formatting information that artificially increases prediction confidence. Furthermore, incorrect labels may break memorization patterns themselves. To ensure meaningful evaluation, the suspect dataset should focus exclusively on text portions guaranteed to appear during training (e.g., question text only), independent of training-specific formatting. As shown in Section 3.3, CoDeC captures the contamination even in cropped samples, so uncertain parts may be safely excluded.

**Impact of data diversity.**  Training datasets consistently yield scores near 100% regardless of their properties. However, unseen dataset scores vary from 0% to 60%, depending on dataset characteristics. CoDeC assumes that additional context from the same distribution provides some additional information, e.g. text type, style, vocabulary, domain knowledge, etc. In large, diverse datasets with unrelated samples, minimal information sharing between datapoints occurs. Hence,

additional context may not improve predictions and can act as random noise, potentially yielding CoDeC scores up to 60%, even for unseen data. Our experiments suggest that this effect may be mitigated using more context samples.

**Larger models use less memorization.** Our main evaluation (Figure 3) included models from 410M to 56B parameters, where training data could be verified. While results remain consistent across architectures and sizes, larger models within families show slightly lower CoDeC scores. This trend intensifies at larger scales. For instance, Llama-Nemotron-Nano 4B exhibits high contamination across multiple benchmarks, while Llama-Nemotron-Ultra 253B maintains all contamination scores below 20%. Although these models share a family, their training processes and data compositions differ; however, these differences alone cannot explain the contamination disparities.

This size-dependent behavior aligns with expectations. CoDeC measures reliance on memorization instead of general knowledge. Larger models possess greater capacity for genuine knowledge acquisition, reducing overfitting to individual datapoints. Consider an illustrative example: when solving algebraic equations, children often rely on pattern memorization, while expert mathematicians apply fundamental skills even for previously encountered problems.

**Adversarial cases.** While CoDeC provides robust contamination scores for most datasets and models, adversarial constructions remain possible. For instance, a dataset containing 1,000 identical (or near-identical) samples produces low CoDeC scores regardless of training exposure. Constructing datasets with artificially high contamination scores is more challenging but possible through combining several unrelated, diverse data sources. See a detailed analysis in Appendix C.7.

These edge cases do not compromise evaluations on standard benchmarks, as genuine datasets remain unaffected by such artificial constructions. Since evaluation data selection remains under user control, models cannot exploit these constructions to bypass detection.

CoDeC applies to any causal model by design. We evaluated over 40 models across various sizes and architectures. One notable exception emerged: GPT-OSS 20B consistently produces contamination scores exceeding 99% across all datasets. Investigation revealed heavy optimization for chat and reasoning tasks that impairs standard language sequence modeling. Even when provided identical text as context, confidence decreases due to persistent attempts to terminate sequences and transition to thinking or chat-like dialogue patterns. Addressing that issue remains for future work.

## A.4 WHY DOES CoDeC WORK?

The central idea behind CoDeC is to measure whether a model has internalized a dataset's specific priors, as their presence is a clear indicator of training data contamination. These priors are not limited to exact string memorization but cover a wide range of cues, such as stylistic patterns, characteristic vocabulary, or common structural templates. While a model may lack the capacity to memorize every single training example, the manifold of these general cues is significantly narrower and can be memorized more easily. CoDeC is designed to detect if the model leverages this learned manifold.

Our approach uses in-context learning as an efficient proxy for finetuning. Consider the standard finetuning process: a model performance improves significantly during the first epoch on a new dataset, with smaller gains in subsequent epochs. Similarly, if we could finetune a model on a target dataset for contamination detection, a contaminated model would improve much more slowly than a non-contaminated one because it has already learned the data distribution. However, performing actual finetuning for every evaluation is computationally prohibitive, especially for large models. CoDeC achieve a similar outcome by using in-context learning, which is a negligible cost. Hence, in essence, CoDeC measures the remaining learning capacity for the target dataset. If the model has already been trained on the data, it has little capacity left to learn, and its performance will not significantly improve when presented with in-context examples from that dataset.

This behavior can be also understood through the lens of the loss landscape. Contamination is often linked to overfitting (Maini et al., 2024), where the model settles into a sharp, narrow local minimum in the loss landscape for the training data. For unseen data, the loss landscape tends to be flatter. Our experiments suggest that in-context learning acts similarly to finetuning with a high learning rate. This makes the process highly sensitive to the local geometry of the loss manifold. For a contaminated sample, the model is already in a steep local minimum. Hence, taking even a single step is likely

to exit this minimum, resulting in worse performance. Conversely, for an uncontaminated sample where the loss manifold is flat, the same step will likely move the model towards a region of higher confidence, improving performance at least slightly.

Finally, CoDeC relates to reference-based MIAs, which often use external models or datasets to calibrate the difficulty of a given sample. CoDeC shares a similar structure but introduces a crucial distinction: it relies on a self-reference approach. Instead of using external data for calibration, CoDeC uses the model own predictions as the baseline. This design choice makes our scores independent of the availability of external resources and ensures that the calibration is based on specifically to the properties and knowledge of the model being investigated.

### A.5    HOW TO INTERPRET CoDeC SCORES?

The CoDeC score is a measure of how much a model's predictions rely on memorized patterns rather than genuine reasoning. Importantly, it does not indicate strict membership of a dataset in the training corpus. Instead, it reflects whether the outputs are predicted based on memorized internal distribution priors – something that can happen if the model was trained on identically distributed or closely related data (e.g. artificially generated). From another perspective, a lower contamination score indicates greater remaining model capacity to learn from the target dataset.

In practice, our experiments reveal two complementary ways to interpret CoDeC scores: absolute evaluation and reference-based comparison.

#### A.5.1    ABSOLUTE SCORE INTERPRETATION

Because CoDeC scores naturally fall between 0% and 100%, they can be compared across datasets and models without model-specific scaling or parameter tuning. Empirically, we found a consistent pattern:

- Scores above 80% were measured for nearly all training datasets we used for experiments, indicating strong contamination evidence.
- Scores below 60% were measured for nearly all unseen datasets. In general, they show no evidence of contamination – the model is likely reasoning based on general knowledge rather than memorization.
- Scores in the 60%–80% range are ambiguous: they may be due to partial contamination or training on related distributions.

Thus, while an absolute threshold (>80%) is a strong indicator, values in the intermediate range require more nuanced analysis.

#### A.5.2    REFERENCE SCORE INTERPRETATION

Absolute scores alone only partially capture variations in dataset properties like diversity. For example, even a non-contaminated model might score relatively high on a broad, highly diverse dataset. To adjust for such effects, the scores should be compared across multiple models on the same dataset.

The most reliable approach is to include at least one model that is known to be non-contaminated with the target dataset (e.g., older model like Pythia). If all models show similar contamination scores, this suggests no substantial memorization specific to any model, but rather a dataset-specific level of the score. However, if a score of the model stands out as an outlier compared to reference models, this strongly suggests higher reliance on memorization.

Choosing reference models of similar size and architecture further helps ensure that differences in scores point to contamination rather than general learning capacity.

#### A.5.3    INFORMED MODEL COMPARISON

Contamination is not inherently negative. For example, models designed to solve advanced math problems are expected to be trained extensively on math datasets for excellence, naturally resulting in higher contamination. This is similar to top IMO competitors who have solved thousands of problems, many from previous competitions, to build their expertise. However, when comparing two

models with similar performance, the one with lower contamination is likely more adaptable and possesses deeper, more general capabilities. Just as a person who excels at IMO without targeted training demonstrates broader intelligence, a model with less reliance on problem-specific priors is usually more capable.

CoDeC measures the remaining learning capacity for the target data distribution, indicating how much room a model has to improve with possible further task-specific tuning. This way, CoDeC allows to understand not only the raw performance of each model, but also the sources of that performance – how much it relies on genuine knowledge and how much on data-specific priors that not necessarily generalize to other related setups.

### A.5.4 PARTIAL CONTAMINATION

Direct inclusion of a benchmark in training is not the only source of contamination. Highly related data – such as paraphrases, augmented samples, identically structured content, or additional examples – can also compromise evaluation, making observed performance less trustworthy. In practice, questions from widely used benchmarks often leak into public sources such as forums, blog posts, and educational material.

As shown in Section 3.3 (Figure 7), CoDeC detects contamination arising from training on related content. Scores should therefore be interpreted in terms of how the training data relates to the target dataset – a relationship that cannot be captured by classical n-gram overlap checks, commonly used in practice (Nguyen et al., 2025).

This explains why some datasets never explicitly seen during training still reach CoDeC scores as high as 60% (see Figure 13). For example, in our experiments the highest-scored unseen datasets were Amazon product reviews, the Colonial Memories book from Project Gutenberg, and Global News articles. Models trained on the Pile dataset have direct exposure to Project Gutenberg and Books3, making Colonial Memories strongly distribution-related even without being part of the training set. Likewise, news and product reviews are common in large web corpora, so high contamination scores for them are not surprising.

Importantly, CoDeC still distinguishes these cases: data directly included in training consistently score near 100%, while related but unseen data get high but noticeably lower scores.

### A.5.5 BEST PRACTICES

For robust contamination assessment, we suggest combining absolute thresholds with reference-based comparisons. High absolute scores (>80%) should be treated as contamination red flags, but significant deviations from other models' CoDeC scores can be equally alarming. By viewing CoDeC scores not as binary labels but as indicators of memorization intensity, the results serve as a useful indicator both for ensuring a fair model comparison on benchmarks, and reliable model quality assessment during training.

We also suggest that, among models with equal benchmark scores, those with lower CoDeC scores should generalize better.

# B    EVALUATION SETUP

To make the main evaluation meaningful, we have to use models for which we know the training data, and prepare additional datasets that were certainly not used for training. Following standard practice, we ensured this exclusion by using data published only after the release of the training datasets, eliminating the risk of data leakage. In each case, we selected a broad range of data types, sources, levels of diversity, and other characteristics to ensure comprehensive coverage of possible data properties in evaluation.

For each dataset, we selected 1 000 random samples for evaluation. If the data source was a continuous stream of text (e.g. book or an article), we split it into 600-characters chunks and treated as a dataset.

Since the selection of both training and unseen datasets is inevitably somewhat arbitrary, we emphasize that the results in this paper reflect all evaluations we have conducted, including internal experiments. Given that the pipeline is simple and lightweight, **we strongly encourage readers to run it themselves and confirm the reliability of CoDeC on datasets of their choice**.

Because most LLMs available today do not disclose their training data, our model choice is necessarily limited. Nevertheless, we selected the following families with full access to their training datasets:

- **Pythia** (Biderman et al., 2023), trained on the Pile dataset (Gao et al., 2021).
- **GPT-Neo** (Black et al., 2021), trained on the Pile dataset.
- **RWKV-4** (Peng et al., 2023), trained on the Pile dataset.
- **OLMo** (Groeneveld et al., 2024), trained on the Dolma dataset (Soldaini et al., 2024).
- **Nemotron-v2** (NVIDIA et al., 2025a), trained on the Nemotron-CC dataset (Su et al., 2024).
- **Nemotron-H** (Blakeman et al., 2025), trained on the Nemotron-CC dataset.

## B.1    TRAINING DATASETS

We used the following sets as training data examples:

**The Pile**

- HackerNews
- Wikipedia
- GitHub
- ArXiv
- DM Mathematics
- Pile CommonCrawl
- PubMed Central
- Full Pile
- Wikipedia Music (a low-diversity subset of Wikipedia containing only music-related articles)
- GitHub Licenses (a low-diversity subset of GitHub containing only license comments)

Since the Pile dataset is not available for direct download, we used the samples provided in the `iamgroot42/mimir` dataset.

**Dolma**

- C4
- Wikipedia
- Pes2o v2
- Reddit v5
- Stack v4

- CommonCrawl head
- CommonCrawl middle
- CommonCrawl tail

**Nemotron-CC**

- Wikipedia
- StackExchange
- ArXiv
- CommonCrawl 2024 (high quality)
- CommonCrawl 2020 (high quality)
- CommonCrawl 2020 (low quality)
- CommonCrawl 2019 (medium quality)
- CommonCrawl 2016 (medium quality)
- CommonCrawl 2014 (low quality)
- CommonCrawl 2013 (high quality)

### B.2 UNSEEN DATASETS

For unseen datasets, we used the following sources:

**Popular benchmarks.** We evaluated the models on the following benchmarks. For each model family, we used those that were published after the training dataset release date to ensure they were unseen:

- For models trained on the Pile: gsm8k, GPQA Diamond, IFEval, HumanEval, FRAMES, AIME 2024, AIME 2025, LiveCodeBench v1, LiveCodeBench v5, BFCL v3, BBQ, RewardBench v1, RewardBench v2, and MATH 500.
- For OLMo models: FRAMES, AIME 2024, AIME 2025, LiveCodeBench v5, BFCL v3, RewardBench v1, and RewardBench v2.
- For Nemotron-H, we used only AIME 2025.

**Project Gutenberg.** We used three books added to Project Gutenberg after April 2025:

- *Colonial Memories*
- *Jibby Jones : A story of Mississippi River adventure for boys*
- *The Corbin necklace*

**Datasets from HuggingFace.**

- `NickyNicky/global-news-dataset`
- `McAuley-Lab/Amazon-Reviews-2023`

**A recent website.**

- An article with Ukrainian conflict updates: `https://www.understandingwar.org/backgrounder/ukraine-conflict-updates`

**Self-created data.**

- A document describing this project.
- A transcript from a meeting.
- Linux syslog file from our computer.
- Internal slack channel log.

## B.3 DATASETS FOR ABLATIONS

For the ablations (Figures 9a, 9b) and experiments with finetuning (Figure 6), we used the following representative 10 datasets:

**Training data (from the Pile)**

- HackerNews
- Wikipedia
- GitHub
- ArXiv
- Full Pile

**Unseen data**

- Colonial Memories (Project Gutenberg)
- Ukrainian conflict updates
- Global News
- gsm8k
- Amazon reviews

## B.4 AUC COMPARISON

The Area Under the Receiver Operating Characteristic Curve (AUC) is a standard metric for evaluating binary classifiers. It measures the probability that a randomly chosen positive example is ranked higher than a randomly chosen negative example, making it threshold-independent and robust to class imbalance. If $S^+$ is the set of scores for positive examples and $S^-$ for negative examples, then:

$$\text{AUC} = \frac{1}{|S^+| \cdot |S^-|} \sum_{s^+ \in S^+} \sum_{s^- \in S^-} \mathbb{1}(s^+ > s^-) + 0.5 \cdot \mathbb{1}(s^+ = s^-)$$

In the context of Membership Inference Attacks (MIAs), AUC is widely used to quantify the classifier's ability to distinguish between training samples (positives) and unseen samples (negatives).

A key difference between our setting and typical MIA evaluations is granularity. In most prior work, contamination scores are computed per sample, and the AUC reflects the quality of sample-level classification. In contrast, our method focuses on the dataset level: we compute a single contamination score per dataset and evaluate the model's ability to classify entire datasets as seen or unseen. This approach better reflects the intended use of CoDeC, which is designed to provide interpretable, aggregate measures of contamination for benchmarks and corpora rather than individual samples.

As shown in Figure 3, CoDeC achieves near-perfect separation between seen and unseen datasets, leading to dataset-level AUC scores close to 100%. This demonstrates that CoDeC is highly effective at capturing training data contamination in a way that aligns with MIA evaluation traditions.

## C  EXTENDED EXPERIMENTS

### C.1  BASELINES

In our analysis, we compare with the following general-purpose baselines:

- **Vanilla loss.** Models are trained to optimize the loss on training examples. Hence, a straightforward way to identify training data is to score the samples based on the average token loss.

- **Min-K%.** Introduced by (Shi et al., 2024), Min-K% builds on the observation that training samples do not have tokens with very low probability. Hence, the criterion computes the score based on the loss measured only on the least-probable tokens in the sequence.

- **Zlib ratio.** Loss-based methods, while effective in some settings, cannot distinguish contaminated data from genuinely easy data, as both result in low model loss. To fix this problem, the Zlib ratio method (Carlini et al., 2021) normalizes the score using sample entropy, estimated as the Zlib compression rate.

### C.2  INFLUENCE OF THE CONTEXT ON MODEL LOGITS

#### C.2.1  CHANGES IN LOGITS WITH CONTEXT

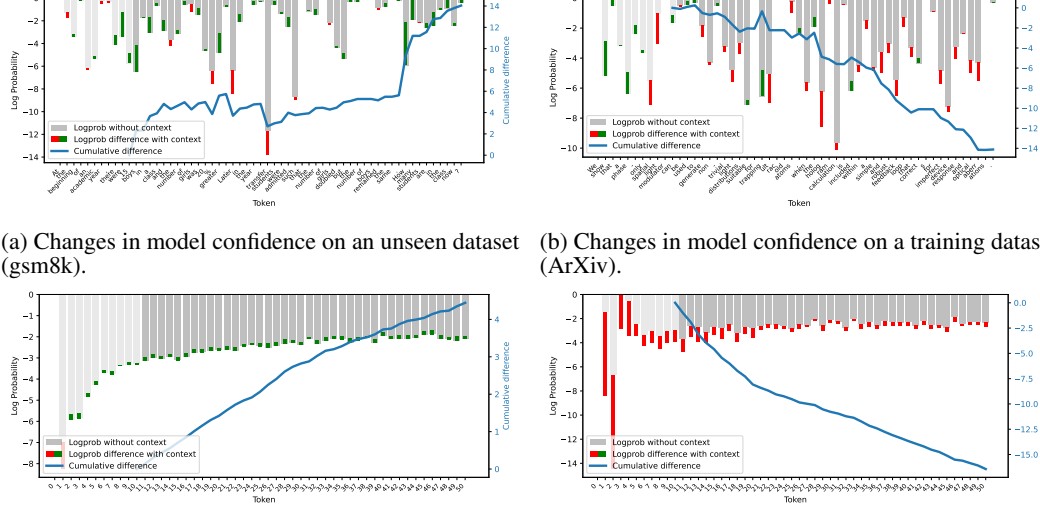

(a) Changes in model confidence on an unseen dataset (gsm8k).

(b) Changes in model confidence on a training dataset (ArXiv).

(c) All samples from GSM8K (unseen), averaged.

(d) All samples from ArXiv (training), averaged.

Figure 10: Change in model log-probabilities with and without context for the Pythia 1.4B model, evaluated on samples from an unseen dataset (gsm8k, 10a) and a training dataset (ArXiv, 10b). The two bottom plots show the differences averaged over the whole dataset.

We observed that models react differently to context depending on whether the dataset was seen during training or not. Specifically, we pass the context by sampling an additional text from the dataset and prepending it to the target sequence (connected with two newlines). Then, we compare the log probabilities for each token in the target sequence with and without the added context. For better stability, we ignore the first 10 tokens of the target sequence to exclude the influence of transitions between texts.

If the dataset from which the samples were taken (both the target text and the context) was seen during training, the log probabilities consistently decrease. However, for unseen datasets, we observe consistently higher estimated probabilities of the correct tokens in the target text. Hence, CoDeC determines whether the samples were seen during training based on whether the cumulative difference is positive or negative. Note that since we check only the sign of the difference, the scale of those fluctuations is not important. Therefore, CoDeC works for each model regardless of its internal properties.

Figure 10a shows the logprob differences for the text "*At the beginning of an academic year, there were 15 boys in a class and the number of girls was 20% greater. Later in the year, transfer students were admitted such that the number of girls doubled but the number of boys remained the same. How many students are in the class now?*" with and without the context text "*Ten adults went to a ball game with eleven children. Adult tickets are $8 each and the total bill was $124. How many dollars is one child's ticket?*". Both texts are sampled from the gsm8k dataset, which was not used for training the Pythia models. Despite the texts having different topics, they share a similar structure; hence, we observe that most tokens in the target text benefit from the added context.

In contrast, Figure 10b shows the logprob differences for the text "*We show that a phase-only spatial light modulator can be used to generate non-trivial light distributions suitable for trapping ultracold atoms, when the hologram calculation is included within a simple and robust feedback loop that corrects for imperfect device response and optical aberrations.*" with and without a context text "*We consider one source of decoherence for a single trapped ion due to intensity and phase fluctuations in the exciting laser pulses. For simplicity we assume that the stochastic processes involved are white noise processes, which enables us to give a simple master equation description of this source of decoherence.*". Both texts are sampled from the ArXiv dataset within the Pile corpus, which was used to train Pythia. Although both texts share a similar structure (both are opening abstracts), both are related to physics, and both use similar vocabulary, nearly all log probabilities decrease when the context is provided due to disrupted memorization.

The described behavior is highly consistent, as demonstrated in our main experiments (Figure 3).

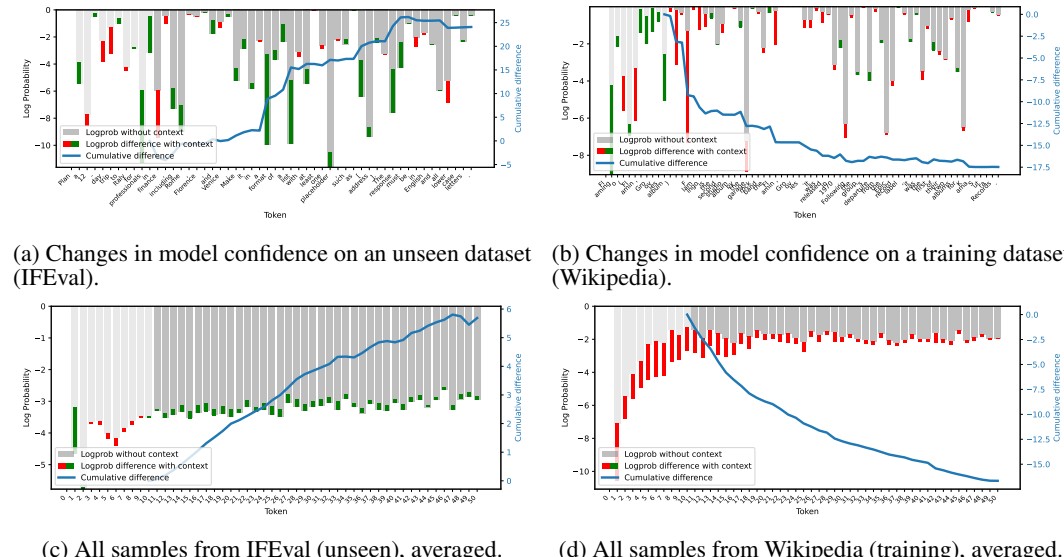

(a) Changes in model confidence on an unseen dataset (IFEval).

(b) Changes in model confidence on a training dataset (Wikipedia).

(c) All samples from IFEval (unseen), averaged.

(d) All samples from Wikipedia (training), averaged.

Figure 11: Additional plots of changes in model log-probabilities with and without context.

### C.2.2 ATTENTION PATTERNS

Figure 12 show how the attention patterns of the model change after exposure to the data. The chart shows attention patterns for the consecutive tokens (each row represents how much the corresponding token attends to previous ones; the lighter the color, the stronger is its reliance). The model (Pythia 410m) is evaluated on 3 random problems from gsm8k (with initially very low contamination) and finetuned for 2 epochs on gsm8k (further training doesn't change much). The contamination rises to 100%. Initially (before training), the model attends to some tokens more and some less (as usual), but generally many tokens are attended to (multiple points in each row are greenish). The attention covers mostly the current question, but somewhat extends also to previous context examples. As the training progresses, the attention patterns sharpen. Tokens that were attended strongly get even stronger, and less important tokens get even weaker. The reliance on the context is nearly completely removed, though some minor influences from the most significant tokens remain visible. It happens

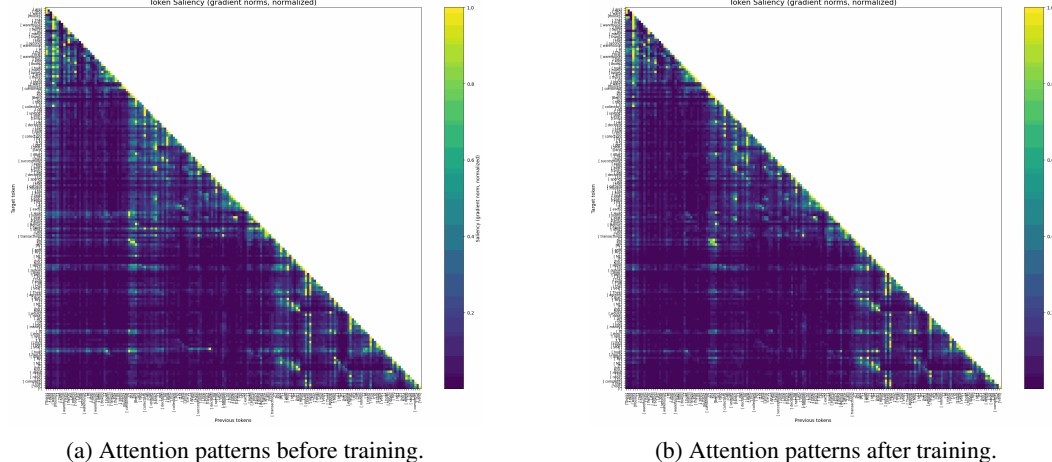

(a) Attention patterns before training.  (b) Attention patterns after training.

Figure 12: Change in attention patterns after training. Each row represents token saliency maps for the next token in the sequence, given the prefix. Yellow/green points indicate high reliance, while dark blue points indicate low reliance. As the training progresses, the reliance polarizes – most predictive tokens become strengthened, less predictive tokens become completely ignored.

mostly during the first steps of finetuning. Later steps strengthen the effect a bit, but generally bring little further changes. This behavior is consistent across different runs and samples.

Non-contaminated models rely on multiple cues, both in the current sequence and in the context. Hence, they can utilize the additional information given. Contaminated models sharply rely on selected tokens – usually these are not spurious words that serve nothing but triggering memorization, but these are genuinely informative tokens that initial model also strongly paid attention to. Just the reliance becomes overly strong and exclusive. Contaminated models tend to ignore the context (note that this is an emergent behavior). Memorization patterns do not sharply transfer from context to other samples; however, their influence is not completely ignored, contributing to at least a bit lower confidence.

## C.3 Per-dataset Score Distribution

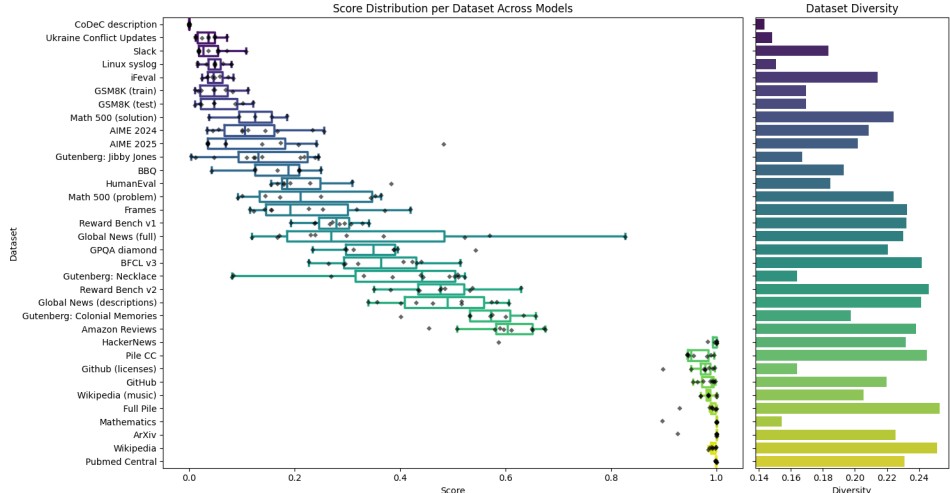

Figure 13: Distribution of CoDeC scores per dataset, computed for models trained on the Pile. The last 10 datasets are parts of the training data. Among the unseen datasets, the largest scores receive those that are more related to the subsets of the Pile.

## C.4 Contamination During Training

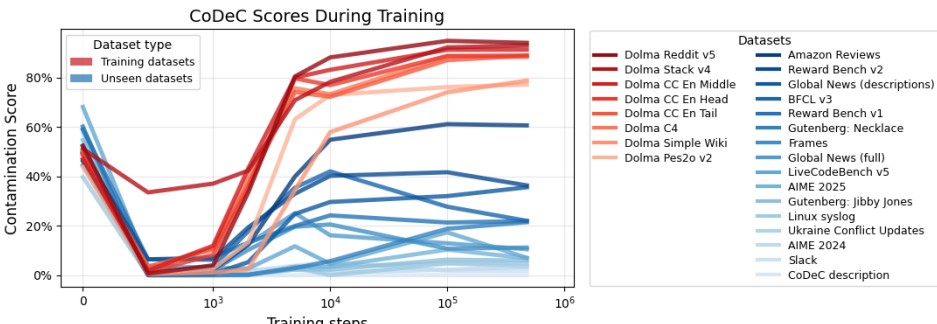

Figure 14: Changes in CoDeC scores over the course of training OLMo 7B. Already after 10k training steps (about 2% of the full training), the contamination scores have nearly converged to their final values. At step 0 all scores oscillate around 50%, as a random model has no ICL skills at all – that is however learned even before step 500, hence all scores immediately go to 0 as the context becomes helpful, but nothing was memorized yet.

## C.5 CONTAMINATION TRANSFER

Contamination transfer from other datasets

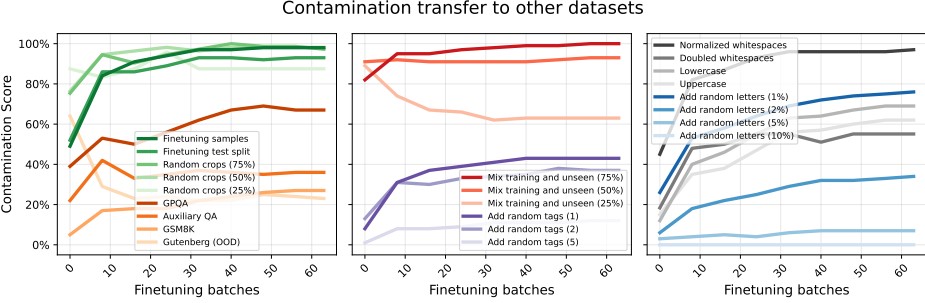

Figure 15: Contamination transfer to MMLU from related distributions, computed for the Pythia 1.4B model.

Contamination transfer to other datasets

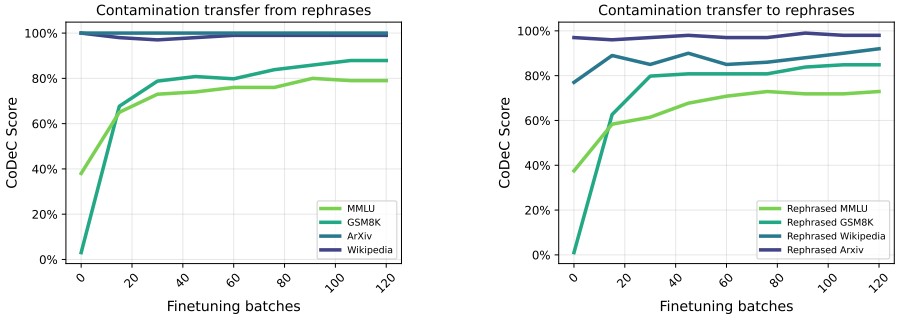

Figure 16: Contamination transfer from MMLU to related distributions, computed for the Pythia 1.4B model.

Figure 17: Contamination transfer between benchmarks and their rephrased versions, computed for the Pythia 1.4B model. Rephrases were obtained using GPT with prompt *"Please rephrase the following text, but keep the meaning the same. Do not change the meaning of the text. Answer with just the rephrased text, no other text or comments."*. In both cases, the contamination is clearly detected.

## C.6 RELEASE-ALIGNED VALIDATION ON ANNUAL BENCHMARKS

We further validate CoDeC in a setting where no training corpora are available. Many benchmarks are released annually; if CoDeC captures reliance on memorized priors, then models whose training cutoff *predates* a given benchmark year should exhibit higher CoDeC scores on those pre-release years than on post-release years.

For each model we define a pre/post partition of benchmark years based on public release timelines (see Table 2). For each (model, year) pair we compute the dataset-level CoDeC score and summarize

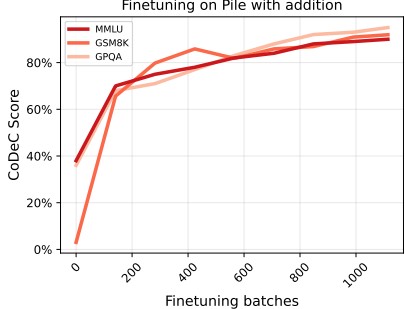 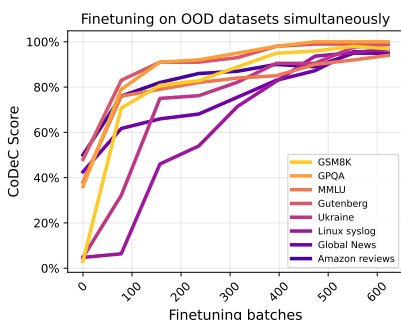

Figure 18: CoDeC scores for a model trained on a broad data mix. In the first experiment (left chart), we mixed multiple subsets of the Pile and the corresponding benchmark, simulating a standard training corpus. In the second experiment (right chart), we mixed multiple unseen datasets and evaluated the progress of CoDeC scores on each component of that mix. In each case, the contamination is clearly detected.

per model across years. Across six models spanning three families, we observe a consistent **Pre > Post** pattern (Figure 19): median CoDeC drops by 5–15 percentage points after the benchmark's release, with per-model Wilcoxon tests significant after FDR correction.

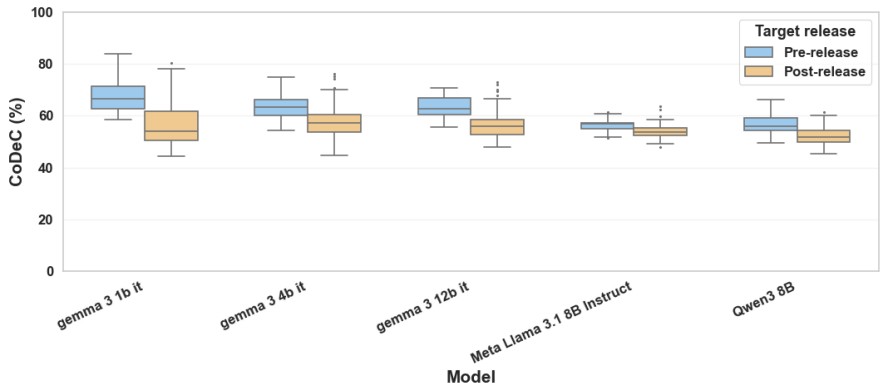

Figure 19: Release-aligned CoDeC scores (%) by model. For each model, benchmark years are grouped into **Pre-release** (blue) vs. **Post-release** (orange) according to the model's training cutoff (see Table 2). Boxes show the distribution of $S_{\text{CoDeC}}$ across years; medians decrease from pre to post for all models, consistent with CoDeC detecting reduced reliance on memorized priors once the benchmark is out-of-distribution relative to the training window.

Table 2: Pre- and post-release IMO years per model (see (HuggingFaceH4 Team, 2024; opencompass Team, 2025)), used for the splits in Figure 19.

| Model | Pre-release years | Post-release years |
|---|---|---|
| Meta Llama 3.1 8B Instruct | $\leq 2022$ | 2023, 2024, 2025 |
| Llama 3.2 3B instruct | $\leq 2022$ | 2023, 2024, 2025 |
| Gemma 3 1B it | $\leq 2023$ | 2024, 2025 |
| Gemma 3 4B it | $\leq 2023$ | 2024, 2025 |
| Gemma 3 12B it | $\leq 2023$ | 2024, 2025 |
| Qwen3 8B | $\leq 2023$ | 2024, 2025 |

## C.7 ADVERSARIAL DATASETS

CoDeC scores provide a reliable and robust contamination estimate, but just like any other metric they are prone to adversarial or degenerate cases. It is possible to construct artificial datasets for

which CoDeC scores will be $0\%$ regardless of the model and its true contamination level. A carefully designed dataset allows to achieve higher contamination scores, though in that case it's much harder to achieve. It means that constructing false positives is not easy even on purpose, which is a desired property.

Observe that it is clearly possible to construct benchmarks that yield universally low or universally high scores regardless of the model evaluated. Likewise, it is possible to construct adversarial datasets for CoDeC. However, such evaluations serve no practical purpose and their existence does not compromise the regular evaluations.

| Adversarial/degenerate datasets | | | | |
| --- | --- | --- | --- | --- |
| Dataset | EleutherAI/pythia-1.4b | RWKV/rwkv-4-430m-pile | allenai/OLMo-1B-hf | Qwen/Qwen3-1.7B |
| One text repeated (ID) | 0 | 0 | 0 | 0 |
| One text repeated (OOD) | 0 | 0 | 0 | 0 |
| One text repeated with noise (ID) | 0 | 0 | 0 | 0 |
| One text repeated with noise (OOD) | 0 | 0 | 0 | 0 |
| Random sequence of words | 0 | 0 | 0 | 0 |
| Adversarial mixture | 41 | 55 | 37 | 57 |

Table 3: CoDeC scores for adversarial and degenerate datasets, carefully crafted to bias evaluations.

### C.7.1 ACHIEVING UNIVERSALLY LOW SCORES

CoDeC scores are a sum of two opposing effects: confidence increase due to the additional information provided in the context, and confidence decrease due to disrupted memorization. If one of those effects is artificially amplified, the scores may become universally lower or higher.

The easiest way to amplify the confidence increase is to inject overwhelming redundancy in the dataset. For instance, if the dataset consists of just a single text copied 100 times, the objective becomes trivial, as the target text is always given precisely in the context. Each model will hence increase its confidence, regardless of whether that text was actually used for training or not. And as shown in Table 3, it is indeed the case.

How severe is that property in practice? As shown in Figure 20, even when we reduce the dataset to as few as 8 distinct samples repeated 16 times each, the CoDeC score remain barely affected. Only reducing the diversity even further visibly reduces the value. In practice, benchmarks exhibiting this level of repetitiveness are a problem themselves.

We also experimented with an extremely degenerate case, in which the dataset consists of random word sequences. The scores equal 0 – even having no related meaning or structure turns out to be a dataset-wide helpful property.

### C.7.2 ACHIEVING UNIVERSALLY HIGH SCORES

While it is very easy to amplify the positive influence, amplifying disruption is much harder. One could think that simply augmenting the previous approach by injecting small random noise (e.g. a single additional random word) would be enough – a model convinced that the task is to copy the context should receive extremely high penalty when failing on the injected word. In practice it is not the case – noisy words indeed penalize the model, but the advantage gained on other tokens always overtakes.

A more effective approach leverages the fact that unrelated context does not successfully improve model's confidence on the target text (see Figure 21). Hence, if we mix several unrelated data sources, the sampled contexts may turn out to be less relevant and fail to increase confidence, resulting in high CoDeC estimates. Indeed, we mixed: IFEval, Linux syslog, GSM8k, and meeting transcript. While

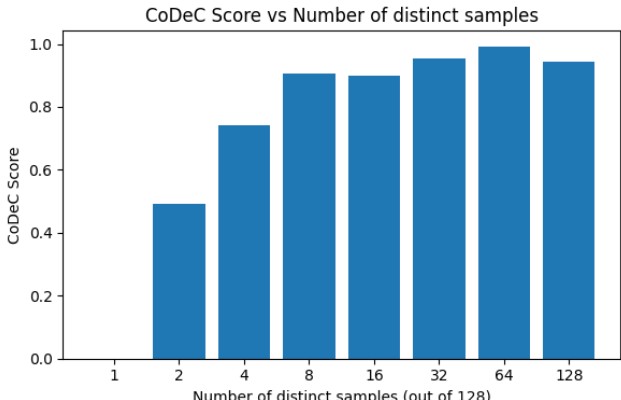

Figure 20: CoDeC scores on repetitive datasets. The x axis is the number of distinct samples in the dataset, indicating repetitiveness factor of $128/x$. The scores are evaluated for the Pythia 1.4b model on Wikipedia.

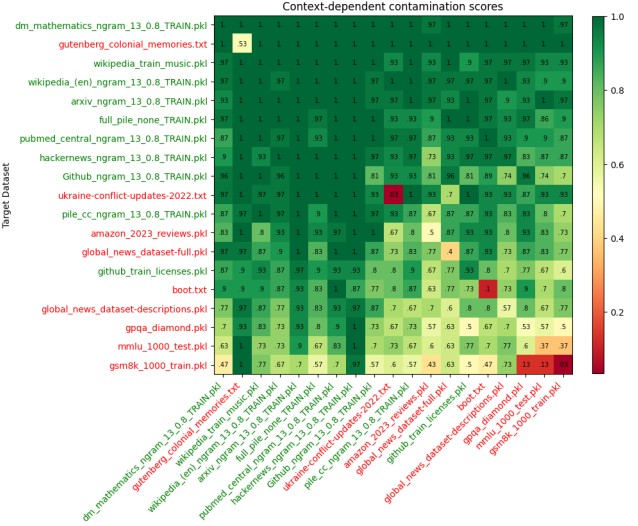

Figure 21: CoDeC ablation: impact of the context distribution. Each cell represents what would be the "ablated contamination score" if for evaluating dataset Y context would be sampled from dataset X, and **not** from X as it is done in CoDeC. Providing unrelated context brings no relevant information, hence the confidence usually drops. Note that those scores are not possible to reproduce within the standard CoDeC pipeline.

each evaluated model gets $< 20\%$ on each of those datasets separately, they get scores of around $50\%$ on the mix. By a careful choice of the subsets, we believe the scores may rise even further.

Such mixes may appear in practice. For instance, MMLU consists of multiple distinct topics, much of which are completely unrelated. However, even in such cases the scores usually do not exceed $50\%$ and remain consistent across different models.

### C.7.3 DEALING WITH ADVERSARIAL CASES

Existence of adversarial datasets is not an issue itself, since the choice of the dataset for evaluating contamination is always on the user side. To counter the influence of unrelated contexts, informed sampling methods (e.g. KNN) may be useful, as they allow to keep the computations within data clusters, making the scores independent of data mixes. Additionally, we note that even though certain data properties may amplify the positive or negative context impact, the main driving forces of CoDeC remain valid, so truly contaminated models still should exhibit considerably higher scores than other models. Hence, it is important to include baseline models in comparison, as suggested in Section 3.5.

## C.8 KNN-BASED CONTEXT SELECTION

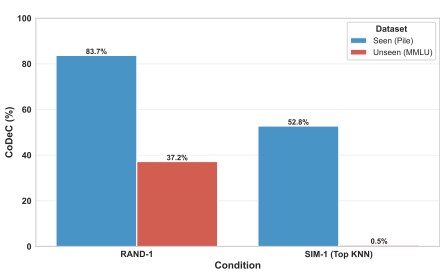
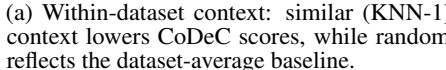
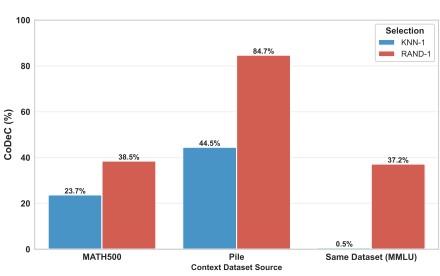

(a) Within-dataset context: similar (KNN-1) context lowers CoDeC scores, while random reflects the dataset-average baseline.

(b) Cross-dataset context: CoDeC drops when context comes from a more similar dataset (MMLU) and remains higher with less similar context (Pile, MATH-500).

Figure 22: CoDeC scores under KNN-based context selection

To better understand how context similarity influences the two effects described in Section 2.2, we conducted additional experiments where the context examples are selected using nearest-neighbor similarity rather than random sampling. This allows us to probe the extremes of the mechanism behind CoDeC: highly similar context should amplify the helpful-information effect, while random context reflects the dataset's average similarity structure. We evaluate both seen and unseen datasets under (1) random context (RAND-1), (2) nearest-neighbor context from the same dataset (SIM-1), and (3) nearest-neighbor context from a different dataset with partial distributional overlap. These experiments are not meant to modify CoDeC, but to provide a controlled validation of its underlying intuition and to clarify how similarity and mismatch influence confidence shifts.

As shown in Figure 22a, choosing similar samples as context reduces the score for unseen MMLU to $0\%$. However, at the same time it reduces the score for seen datasets as well, leaving the gap mostly unchanged. Clearly, choosing more similar contexts amplifies the positive influence. To preserve practicality of the criterion, the magnitude of that intervention should be carefully chosen to maximize the separation between seen and unseen datasets.

Similarity-based selection has the potential to mitigate adversarial data mixtures. Combining multiple unrelated datasets amplifies the scores due to less relevant contexts being sampled, but choosing contexts among the most similar examples can successfully mitigate this issue.

Incorporating KNN-based context sampling introduces additional hyperparameters to the pipeline – embedding model, similarity metric, how to leverage similarity, etc. Therefore, to preserve the simplicity of the pipeline we decided to choose the simplest option of sampling the context uniformly, as it is sufficient to achieve good results and leaves less decisions for the user. We leave exploring similarity-based context sampling for future work.

# D  BROADER EVALUATION

Experiments on models with known training data, presented in Section 3, confirmed the reliability of CoDeC. With confidence in its validity, we subsequently applied CoDeC to a broad suite of recent models and analyzed their contamination scores across widely used benchmarks.

Table 4 summarizes the CoDeC scores for various models on popular benchmarks. Most models scored below our high-contamination threshold of $80\%$, suggesting that none were directly trained on benchmark data. However, several scores approach this threshold (e.g., Phi-4-reasoning-plus on AIME 2024, Devstral-Small on AIME 2025, Qwen3-30B-A3B-Thinking on MATH 500, Nemotron-Nano-8B on Reward Bench v1, etc). Such results may indicate partial leakage of benchmark data into training corpora, or extensive training on closely related data (e.g., synthetically generated MMLU questions). In either case, high scores suggest benchmark accuracy may be inflated by memorization rather than genuine reasoning.

A notable case is Qwen 2.5, which scored $100\%$ on the GSM8K training set but only $27\%$ on its test set. According to the technical report (Yang et al., 2024), GSM8K's training set was deliberately included in training, but questions overlapping heavily with the test set were excluded. CoDeC detected this difference, validating its effectiveness. Additionally, it demonstrates that even simple overlap detection strategies can be successful to avoid contamination if applied carefully.

As discussed earlier, models with outlier scores on a given benchmark warrant higher attention. We observe clear examples in IFEval, Hellaswag, and RewardBench v1, where unusually high scores of single models suggest contamination. Hence, accuracy results on such benchmarks should be interpreted with caution for those models.

For GSM8K and Hellaswag, we evaluated both training and test sets. Scores were generally similar across both splits, except for Qwen and related models due to deliberate decontamination. This suggests that other model families, if they directly or indirectly use GSM8K training data, made little or no effort to remove overlaps with the test set.

We also computed CoDeC scores for both problems and solutions in the MATH-500 benchmark. Several models scored high on problem statements but all scores on solutions are rather low.

Our evaluation suite includes diverse model types and architectures: base models, instruction-tuned models, reasoning-optimized models, mixture-of-experts models, and both older and recent releases. Since CoDeC is model-agnostic, it produces meaningful scores regardless of model category. Variants within the same family (e.g., Llama-3.1-8B vs. Llama-3.1-8B-Instruct) typically produce very similar scores, suggesting all variants are equally suitable for CoDeC evaluation.

We observed that the largest models tend to have much lower contamination scores. For instance, Llama-Nemotron 8B appears somewhat contaminated with multiple benchmarks (MMLU-Pro, RewardBench v1, AIME 2024, etc.), whereas Llama-Nemotron 235B scores between $0\%$ and $20\%$ across all evaluations. Larger models are known to generalize better and thus rely less on memorization. We conjecture that, regardless of overlap between training data and benchmarks, large models depend more on generalization, resulting in lower contamination scores. As an analogy, a child shown the solution to a quadratic equation may attempt to memorize it, being unable to solve it otherwise; an experienced mathematician, however, would not memorize the example since solving it is more straightforward given their knowledge.

| Model | AIME 2024 | AIME 2025 | BBQ | BFCL v3 | FRAMES | GPQA Diamond | gsm8k (test) | gsm8k (train) | hellaswag (test) | hellaswag (train) | HumanEval | IFEval | LiveCodeBench v1 | LiveCodeBench v5 | MATH 500 (problem) | MATH 500 (solution) | MMLU-Pro | RewardBench v1 | RewardBench v2 |
|---|---|---|---|---|---|---|---|---|---|---|---|---|---|---|---|---|---|---|---|
| deepseek-ai/DeepSeek-R1-Distill-Llama-8B | 16 | 20 | 16 | 37 | 25 | 54 | 23 | 25 | 8 | 7 | 27 | 8 | 57 | 58 | 31 | 11 | 53 | 38 | 58 |
| deepseek-ai/DeepSeek-R1-Distill-Qwen-14B | 11 | 20 | 8 | 45 | 12 | 51 | 26 | 60 | 7 | 9 | 32 | 7 | 58 | 52 | 36 | 7 | 52 | 33 | 52 |
| deepseek-ai/DeepSeek-V2 | - | 6 | 0 | 3 | 4 | 19 | 10 | 11 | 1 | 2 | 14 | - | 24 | 19 | 16 | 7 | 33 | 20 | 23 |
| EleutherAI/pythia-1.4b | 2 | 3 | 0 | 24 | 8 | 28 | 0 | 1 | 3 | 3 | 16 | 2 | 7 | 9 | 4 | 12 | 29 | 21 | 36 |
| EleutherAI/pythia-12b | 0 | 3 | 4 | 15 | 9 | 25 | 6 | 5 | 11 | 3 | 28 | 2 | 31 | 30 | 5 | 14 | 30 | 19 | 32 |
| google/gemma-3-12b-it | 3 | 0 | 4 | 17 | 6 | 13 | 11 | 10 | 3 | 2 | 3 | 1 | 0 | - | 15 | 6 | 19 | 17 | 22 |
| google/gemma-3-12b-pt | 6 | 10 | 0 | 2 | 7 | 24 | 11 | 13 | 1 | 2 | 11 | 0 | 33 | 27 | - | 5 | 40 | 18 | 29 |
| google/gemma-3-27b-it | 3 | 0 | 12 | 25 | 5 | 15 | 10 | 11 | 3 | 3 | 1 | 2 | 0 | - | 17 | 6 | 14 | 19 | 31 |
| google/gemma-3-27b-pt | 3 | 17 | 0 | 3 | 6 | 23 | 12 | 14 | 0 | 1 | 11 | 0 | 22 | 18 | 10 | 5 | 39 | 16 | 28 |
| google/gemma-3-4b-it | 2 | 10 | 12 | 17 | 7 | 16 | 9 | 8 | 3 | 4 | 11 | 5 | 0 | 0 | 13 | - | 18 | 19 | 23 |
| google/gemma-3-4b-pt | 2 | 0 | 4 | 6 | 7 | 32 | 12 | 15 | 1 | 2 | 15 | 0 | 35 | - | - | 3 | 44 | 22 | 33 |
| meta-llama/Llama-3.1-70B | 11 | 13 | 0 | 6 | 6 | 24 | 19 | 19 | 3 | 4 | 63 | 0 | 15 | 15 | 17 | 17 | 38 | 19 | 23 |
| meta-llama/Llama-3.1-8B | 41 | 62 | 8 | 6 | 6 | 32 | 19 | 20 | 3 | 4 | 39 | 0 | 18 | 16 | 27 | 37 | 43 | 20 | 28 |
| meta-llama/Llama-3.1-8B-Instruct | 41 | 41 | 8 | 16 | 8 | 39 | 24 | 24 | 3 | 3 | 17 | 15 | 53 | - | - | - | 47 | 29 | 38 |
| microsoft/phi-4 | 75 | 51 | 8 | 27 | 19 | 28 | 46 | 37 | 70 | 82 | 12 | 1 | 11 | 7 | 74 | 39 | 48 | 31 | 29 |
| microsoft/Phi-4-mini-instruct | 52 | 55 | 12 | 39 | 20 | 40 | 58 | 73 | 72 | 73 | 26 | 20 | 30 | 28 | 61 | 44 | 53 | 29 | 50 |
| microsoft/Phi-4-reasoning-plus | 84 | 69 | 12 | 40 | 18 | 36 | 47 | 66 | 64 | 81 | 11 | 8 | 40 | 41 | 80 | 26 | 55 | 38 | 53 |
| mistralai/Devstral-Small-2507 | 60 | 82 | - | 9 | 7 | 28 | - | 15 | 2 | 2 | 9 | - | 20 | 17 | - | - | 40 | 22 | 33 |
| mistralai/Magistral-Small-2507 | 37 | 58 | 8 | 18 | 7 | 28 | 13 | - | 3 | 2 | 28 | - | 22 | 21 | 32 | 5 | 41 | 26 | 40 |
| mistralai/Ministral-8B-Instruct-2410 | 13 | 27 | - | 18 | 11 | 40 | 13 | 15 | 3 | 4 | 16 | - | 11 | - | 26 | 21 | 45 | 30 | 54 |
| mistralai/Mistral-7B-v0.1 | 7 | 6 | 8 | 6 | 6 | 33 | 8 | 9 | 1 | 2 | 18 | 0 | 21 | 20 | 13 | 5 | 39 | 25 | 36 |
| mistralai/Mistral-Large-Instruct-2411 | 48 | 58 | 4 | - | 6 | 31 | 18 | 25 | 3 | 3 | 8 | 4 | - | - | 68 | 34 | 47 | 27 | 50 |
| mistralai/Mixtral-8x7B-v0.1 | 30 | 24 | 4 | 5 | 7 | 26 | 11 | 11 | 2 | 2 | 25 | 0 | 15 | 14 | 17 | 7 | 32 | 22 | 30 |
| nvidia/Llama-3.1-Nemotron-Nano-8B-v1 | 66 | 62 | 37 | 46 | 37 | 56 | 43 | 42 | 10 | 12 | 32 | 57 | 38 | 35 | 61 | 21 | 71 | 76 | 58 |
| nvidia/Llama-3_1-Nemotron-Ultra-253B-v1 | 0 | 0 | 8 | 12 | 5 | 6 | 13 | 12 | 1 | 1 | 1 | 3 | 2 | 2 | - | - | 19 | 17 | 22 |
| nvidia/Llama-3_3-Nemotron-Super-49B-v1 | 32 | 34 | 12 | 38 | 10 | 37 | 49 | 49 | 6 | 8 | 9 | 19 | 39 | 35 | 30 | 19 | 49 | 35 | 65 |
| nvidia/Mistral-NeMo-Minitron-8B-Instruct | 35 | 34 | 4 | 15 | 10 | 25 | 34 | 36 | 4 | 2 | 7 | 8 | 6 | 7 | 41 | 45 | 31 | 19 | 42 |
| nvidia/Nemotron-H-47B-Base-8K | 26 | 3 | 0 | 8 | 5 | 11 | 12 | 38 | 2 | 3 | 4 | 4 | 2 | - | - | - | 29 | 22 | 24 |
| nvidia/Nemotron-H-56B-Base-8K | 20 | 10 | 0 | 8 | 5 | 7 | 15 | 50 | 2 | 3 | 8 | 1 | 1 | - | - | - | 30 | 21 | 20 |
| nvidia/Nemotron-H-8B-Base-8K | 38 | 31 | 8 | 19 | 8 | 37 | 13 | 75 | 3 | 3 | 21 | 6 | 35 | 35 | - | - | 49 | 29 | 39 |
| nvidia/OpenReasoning-Nemotron-14B | 17 | 20 | 12 | 20 | 20 | 38 | 29 | 30 | 5 | 5 | 14 | 16 | 10 | 11 | - | - | 34 | 23 | 27 |
| nvidia/OpenReasoning-Nemotron-32B | 17 | 13 | 12 | 24 | 21 | 33 | 19 | 27 | 11 | 12 | 40 | 14 | 56 | 63 | - | - | 40 | 36 | 38 |
| nvidia/OpenReasoning-Nemotron-7B | 16 | 13 | 25 | 24 | 20 | 37 | 42 | 38 | 4 | 3 | 45 | 22 | 70 | 70 | - | - | 38 | 29 | 34 |
| Qwen/Qwen2.5-1.5B | 60 | 10 | 8 | 25 | 16 | 46 | 24 | 99 | 5 | 6 | 60 | 5 | 23 | 13 | 57 | 20 | 49 | 34 | 48 |
| Qwen/Qwen2.5-1.5B-Instruct | 57 | 27 | 12 | 40 | 17 | 60 | 27 | 100 | 4 | 6 | 54 | 21 | 43 | 24 | 60 | 16 | 60 | 38 | 63 |
| Qwen/Qwen2.5-14B | 64 | 10 | 4 | 17 | 13 | 25 | 16 | 70 | 3 | 3 | 54 | 1 | 13 | 8 | 71 | 12 | 39 | 30 | 39 |
| Qwen/Qwen2.5-72B | 61 | 6 | 4 | 15 | 11 | 14 | 11 | 45 | 3 | 3 | 48 | 2 | 5 | 6 | 68 | 8 | 37 | 29 | 36 |
| Qwen/Qwen2.5-7B | 62 | 6 | 8 | 19 | 15 | 31 | 20 | 68 | 7 | 3 | 48 | 3 | 10 | 7 | 70 | 14 | 40 | 30 | 40 |
| Qwen/Qwen3-14B | 7 | 0 | 8 | 34 | 13 | 32 | 28 | 54 | - | 5 | 57 | 7 | 4 | 4 | 14 | 6 | 37 | 34 | 48 |
| Qwen/Qwen3-30B-A3B-Instruct-2507 | 55 | 17 | 12 | 43 | 12 | 23 | 57 | 96 | 6 | 8 | 64 | 19 | 51 | 41 | 77 | 29 | 46 | 35 | 60 |
| Qwen/Qwen3-30B-A3B-Thinking-2507 | 40 | 37 | 8 | 39 | 14 | 37 | 53 | 93 | 7 | 8 | 67 | 34 | 31 | 28 | 80 | 24 | 52 | 39 | 63 |
| Qwen/Qwen3-4B-Thinking-2507 | 15 | 31 | 16 | 29 | 17 | 28 | 45 | 82 | 6 | 5 | 58 | 15 | 31 | 33 | 46 | 13 | 45 | 31 | 50 |
| Qwen/Qwen3-8B | 2 | 6 | 8 | 29 | 15 | 30 | 14 | 21 | 5 | 6 | 42 | 7 | 8 | 10 | - | 5 | 35 | 30 | 44 |
| Qwen/QwQ-32B | 7 | 3 | 8 | 36 | 13 | 45 | 3 | 56 | 13 | 12 | 54 | 9 | 48 | 37 | 22 | 5 | 42 | 31 | 59 |

Table 4: CoDeC scores for widely used models across a range of popular benchmarks. Scores are expressed as percentages. Where available, both train and test sets are included in the table.

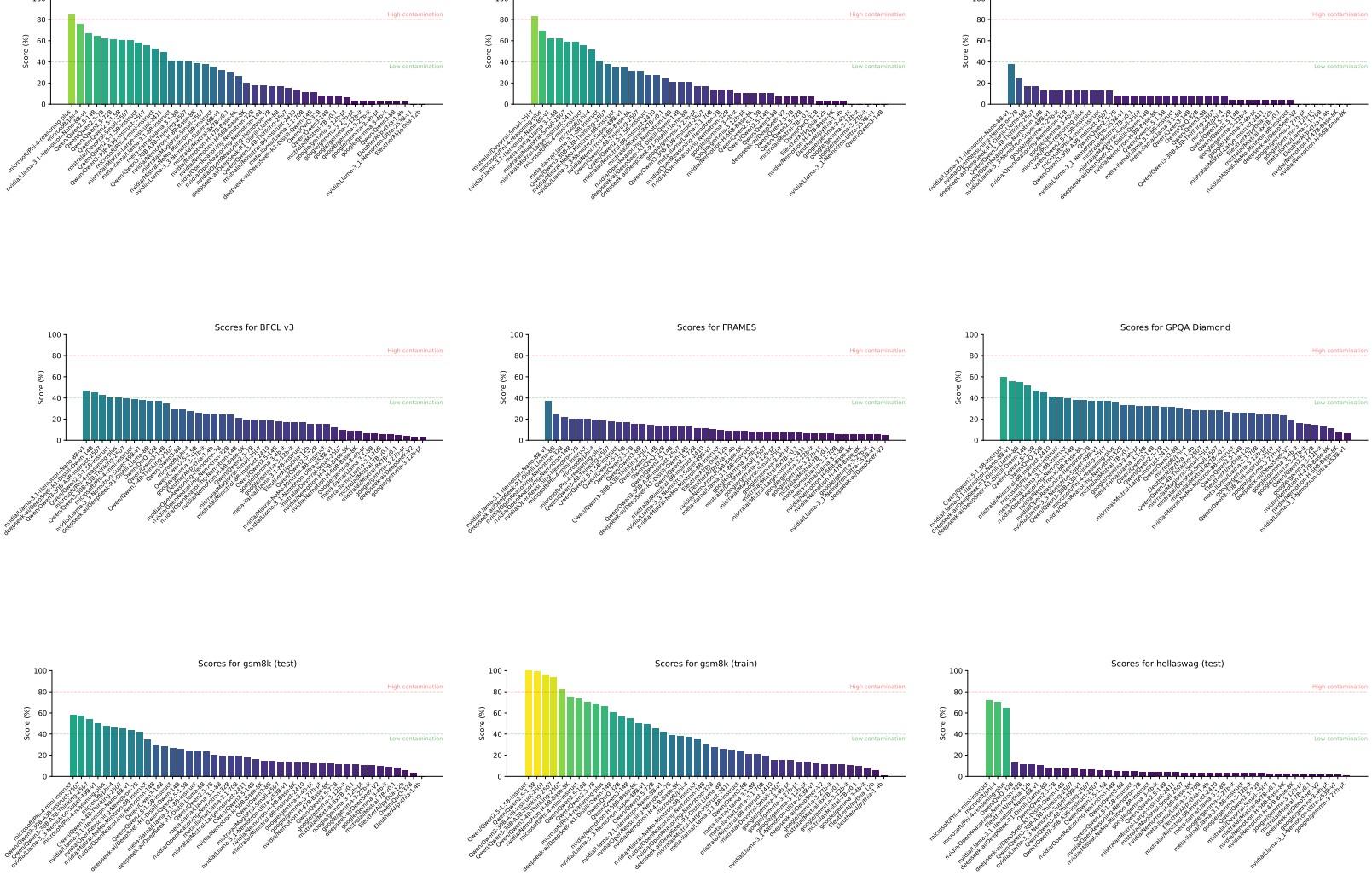

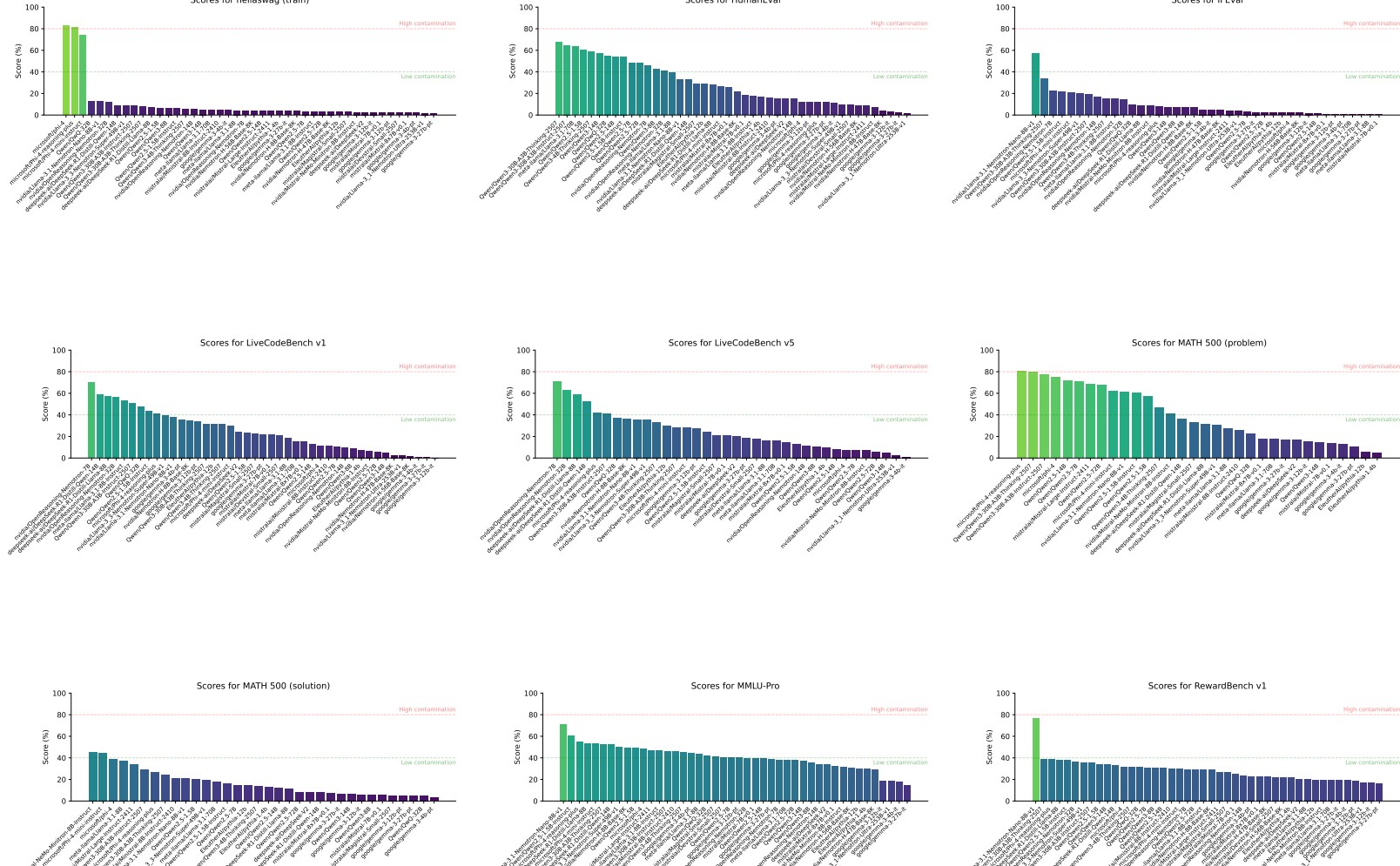

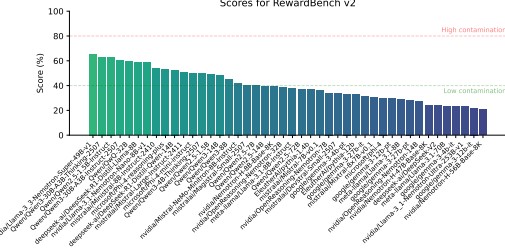

