# OpenReview forum: "Detecting Data Contamination in LLMs via In-Context Learning"
_ICLR.cc/2026/Conference — ICLR 2026 Poster_

### Official Review · Reviewer_7wL6 · 2025-10-29

**Soundness:** 3
**Presentation:** 3
**Contribution:** 3
**Rating:** 6
**Confidence:** 4

**Summary:**

The authors propose a method called CoDeC (Contamination Detection via Context) for detecting dataset-level contamination. The key idea is simple yet effective, for unseen data ICL examples will boost its confidence, so Delta (=ICL - no ICL) would be positive, but for trained data the ICL examples could be confusing hence result in negative Delta. The authors conducted wide range of experiments to support their method.

**Strengths:**

1. Clean and simple idea. I always appreciate simple ideas that work, and this paper's idea is novel and effective.
2. Extensive experiments. The authors conducted experiments on a wide range of model families (even for RWKV ones), all with strong experiment results.
3. Well-written and of importance to the field. The writing is clear and the topic is important to the community.

**Weaknesses:**

I have 2 key questions:
1. Does CoDeC work when you rephrase the question? (either train on rephrased versions then test on original, or train on original then test on the rephrased) I think it's getting widely known that many contamination cannot detect contamination unless it's under exact same phrasing, and arguably the value is low if a method cannot work for general use cases.
2. Did you try training with a mix of other datasets in your experiments? i.e. not only the contamination test set, but with general corpus? This might improve models general ICL ability and render CoDeC ineffective, and people usually train their model this way instead of only using the test set.

**Questions:**

1. Does model's capability have a correlation with the method effectiveness? Could you also report the general eval metrics (for benchmarks other that the contaminated data) of the models before and after finetuning?

---

> ### Author Response · Authors · 2025-11-18
>
> Dear Reviewer 7wL6,
>
> We thank you for your feedback – we share your enthusiasm for simple ideas that work. Below, we address your questions and comments. We would be happy to continue the discussion on any point in more detail.
>
> > Does CoDeC work when you rephrase the question?
>
> Yes. In additional experiments added for the rebuttal (Section E.3, Figures 19a-19b), we show that training solely on rephrased samples is sufficient for CoDeC to detect high contamination, and conversely, training on original data increases contamination scores for rephrased samples. Thank you for suggesting us this experiment.
>
> CoDeC is not dependent on exact phrasing. As demonstrated in Section 3.3 and Appendix C.5, it detects contamination even after training on augmented versions of the dataset (random cropping, noise injection, whitespace/casing changes, etc.).
>
> Furthermore, when finetuning on only half of a dataset, the model exhibits high contamination scores on the unseen half (see Figure 7). This behavior is consistent – for each model trained on the Pile (Pythia, GPT-neo, RWKV) we observed that their CoDeC scores with train splits of Pile are nearly identical as their scores on test splits. Moreover, Figure 4 shows that after only 10k training steps of OLMo (having seen ~2% of each dataset) CoDeC correctly estimates contamination levels for both seen and unseen data.
>
> This robustness arises because CoDeC scores are computed via pairwise sample comparisons, emphasizing distribution‑level priors rather than sample-level. This makes CoDeC broadly applicable for reliable contamination detection, even under rephrasing or other data transformations.
>
> > Did you try training with a mix of other datasets in your experiments?
>
> In Appendix C.5, we show that a model becomes highly contaminated with MMLU even when mixed with another dataset at varying fractions (25%-75%). Following your suggestion, we conducted additional evaluations (Section E.3 of the rebuttal experiments) examining:
> - Training on combined Pile subsets with the addition of a single benchmark dataset.
> - Training on combinations of multiple unseen datasets.
>
> In both cases, contamination levels easily approach nearly 100%, confirming CoDeC sensitivity and effectiveness.
>
> We note that our finetuning experiments are intended as supplementary validation and do not fully replicate full-scale training pipelines. The primary evidence for CoDeC robustness remains our main validation (Figure 3), which evaluates models trained through complete, realistic workflows. In practice, detecting real-world benchmark contamination is analogous to detecting contamination from portions of the training set.
>
> > Does model's capability have a correlation with the method effectiveness? Could you also report the general eval metrics (for benchmarks other that the contaminated data) of the models before and after finetuning?
>
> We appreciate this insightful question. A key property of CoDeC is that its contamination scores are largely independent of overall model capabilities. For example, in our training‑progress experiment (Figure 4), CoDeC detects the same contamination level after only 10k training steps as in the fully trained model, despite the early checkpoint being substantially less capable.
>
> This consistency arises because CoDeC measures the interplay between the positive influence of relevant context and the negative influence of memorization, both of which are present in models with even basic in‑context learning ability. Modern models – even relatively weak ones – generally meet this requirement.
>
> Clearly, after finetuning the model reaches around 100% accuracy on the training data. However, we use our finetuning setup as a supplementary validation, hence it does not reproduce the real training closely. For a fully meaningful general benchmark evaluation, we should more closely simulate that setup.

---

> ### Author Response · Authors · 2025-11-30
>
> Dear Reviewer 7wL6,
>
> We would like to thank you for your very positive and insightful feedback, particularly for suggesting the experiment with synthetic rephrasing of the tested samples, which turned out to be especially interesting -- it will definitely strengthen the analysis of CoDeC.
>
> Also, we would like to thank you once again for the positive reception of our research -- we share your enthusiasm for simple ideas that work. We hope that our responses have addressed all your remaining concerns and have further convinced you of the high quality of our method.

---

### Official Review · Reviewer_sLbA · 2025-10-31

**Soundness:** 2
**Presentation:** 3
**Contribution:** 3
**Rating:** 4
**Confidence:** 4

**Summary:**

This paper introduces Contamination Detection via Context (CoDeC) — a simple, scalable method for detecting and quantifying training data contamination in large language models (LLMs) using in-context learning dynamics.


The key idea

* When given in-context samples from a dataset it has not seen during pre-training, an LLM tends to increase its confidence on the target sample.
* When the dataset has been seen (i.e., contaminated), additional context disrupts memorization and reduces confidence.

The method computes a contamination score by measuring the fraction of samples for which adding in-context examples decreases the model’s confidence.


Main contributions

* Proposes CoDeC — a model- and dataset-agnostic contamination detection method requiring only gray-box access to logits.
* Demonstrates near-perfect separation (AUC ≈ 99.9%) between seen and unseen datasets across various models (Pythia, GPT-Neo, RWKV, OLMo, Nemotron).
* Shows robustness to dataset diversity and training stage, and scalability to large benchmarks.
* Highlights how contamination transfers across related datasets and how finetuning affects CoDeC scores.
* Positions CoDeC as an interpretable, efficient alternative to classical membership inference methods.

**Strengths:**

Originality

* While prior work relies heavily on membership inference, loss-based calibration, or reference models, CoDeC offers a novel formulation by exploiting in-context learning behavior as a contamination signal.
* The idea is conceptually elegant — turning a standard model property (ICL behavior) into a contamination test.

Quality

* The experiments are extensive: multiple models (spanning architectures and sizes), datasets (training vs unseen), and baselines.
* Ablation studies are included (context size, dataset size) and finetuning experiments provide additional validation.
* Clear definition of the contamination score and rationale for why it works.

Clarity

* The paper is very well written, with clean exposition of the problem statement and intuition.
* Figures are informative and minimalistic, making the method easy to grasp.
* The pipeline (Fig. 1) clearly communicates the method steps.

Significance

* Data contamination detection is increasingly critical for fair evaluation of LLMs.
* A method that is scalable, interpretable, and does not require access to training data or reference models is very valuable.
* The simplicity and efficiency of CoDeC make it applicable at scale, which is practically important.

**Weaknesses:**

Experiment rigor to test the generality of the findings

* The paper claims with the help of CMA that LLMs
* Similar unsubstantiated claims appear in Section 2.2 (Key Idea), e.g., that adding in-context samples usually improves confidence for unseen datasets — no citations or empirical backing are provided there.

Limited theoretical grounding

* While intuition is discussed in depth (Section 2.5), the theoretical explanation remains qualitative.
* Formal guarantees or bounds on false positive/negative rates for contamination detection would strengthen the contribution.

Evaluation scope

* Although contamination transfer across related datasets (e.g., MMLU) is explored, adversarial or edge cases (e.g., near-identical but unseen datasets, noisy mixtures) are only lightly discussed.
* The method might conflate contamination with related distribution overlap, as the authors acknowledge, but do not quantitatively evaluate how severe this is.

Novelty relative to related work

* Some elements overlap conceptually with loss-based or entropy-based detection. Positioning CoDeC more clearly in terms of unique advantages and trade-offs would help.

**Questions:**

Empirical clarification

* Can the authors provide empirical evidence or references to support the claims in Section 2.2 and Section 3.5 regarding confidence shifts and generalization ability?
* Can they quantify how CoDeC correlates with generalization metrics across models with similar benchmark accuracy?

Contamination vs distributional similarity

* How robust is CoDeC to partial contamination or to unseen but stylistically similar datasets?
* Could the authors provide additional results where datasets are synthetically perturbed to vary similarity to training data?

Thresholding strategy

* The paper mentions thresholding and comparison against other models. Could the authors elaborate on practical guidelines for selecting these thresholds in real-world evaluation scenarios?

Theoretical properties

* Do the authors have any theoretical guarantees (e.g., consistency or error bounds) on CoDeC scores? Or atleast outline what such analysis might require?

Reproducibility and usage

* How sensitive is the method to the number of in-context samples and the randomness of context selection?
* Would deterministic context selection (e.g., nearest neighbors) further improve stability?

---

> ### Author Response · Authors · 2025-11-18
>
> Dear Reviewer sLbA,
>
> Thank you for your thorough and constructive feedback. We are glad that you recognized the novelty and elegance of our method. Below, we address your questions and comments, and we would be happy to continue the discussion if any points remain unclear.
>
> > Can the authors provide empirical evidence or references to support the claims in Section 2.2 and Section 3.5 regarding confidence shifts and generalization ability?
>
> This is a very important question. In the attached experiments (additional rebuttal experiments, Section E.4), we plot how empirically the logits change with or without the context, both for individual examples and as aggregated trends. We also analyze the attention maps to show how memorization patterns change after exposure to the data. We added selected visualizations to the Introduction to improve the explanation of the key ideas.
>
> The main idea of CoDeC is based on the property that a context sampled from a dataset has twofold influence on predictions: (1) it improves confidence as it brings additional information about the dataset distribution, (2) it decreases confidence as it disturbs the learned memorization patterns. The final difference in logits with and without the context given is the sum of those effects.
>
> In case of unseen datasets there is little to no memorization involved, hence mostly the positive factor makes an impact. Conversely, if the model has seen the given dataset during training, it has already internalized the priors, so the positive influence is minimal, but it can be severely disturbed because it learned to memorize those sequences without context. The attached empirical analysis confirms those claims on the token-level. Additionally, we note that the effectiveness of CoDeC shown in our main validation experiment (Figure 4) is a further empirical proof of those claims.
>
> We attach a few references reinforcing our key insights:
> In-context examples improving confidence on unseen tasks
> - Brown et al., 2020; Language models are few-shot learners.
> - Min et al., 2022; Silo language models: Isolating legal risk in a nonparametric datastore
> Memorized sequences disrupted by added context
> - Razeghi et al., 2022; Impact of Pretraining Term Frequencies on Few-Shot Reasoning
> - Carlini et al., 2022; Extracting training data from large language models.
>
> They will be subsequently added to Section 2 to improve the clarity of our explanations. Thank you for bringing this to our attention.
>
> > Can they quantify how CoDeC correlates with generalization metrics across models with similar benchmark accuracy?
>
> While generalization is inherently difficult to measure objectively, we explored connections between CoDeC scores and LM arena rankings as a proxy for generalization. One example is shown below:
>
> ```
> Model               | Math500 acc | Math500 contamination | AIME 25 acc | AIME 25 contamination | LM arena math leaderboard position
> Phi-4 (14B)         | 81%         | 74%                   | 18%         | 51%                   | 127
> gemma3-12b-it (12B) | 85%         | 15%                   | 18%         | 0%                    | 77
> ```
>
> These two models have similar sizes and comparable benchmark accuracies, yet their contamination scores differ substantially. Consistent with this, Gemma ranks significantly higher in the LM arena math leaderboard, which we use here as a proxy for generalization, suggesting that lower contamination scores may be associated with better real‑world generalization among comparable models. While this is only one illustrative case, the magnitude of the difference supports the potential of CoDeC scores to complement accuracy metrics in evaluating models.

---

> ### Author Response · Authors · 2025-11-18
>
> > How robust is CoDeC to partial contamination or to unseen but stylistically similar datasets?
> > Could the authors provide additional results where datasets are synthetically perturbed to vary similarity to training data?
>
> This is a very good question. CoDeC is designed to detect contamination from related datasets, not only exact test samples. As shown in Section 3.3 and Appendix C.5, training on synthetically perturbed variants (e.g., random cropping, noise injection, whitespace changes, casing modifications, etc) reliably increases contamination scores on the original dataset.
>
> Following your suggestion, we extended our contamination transfer experiments to cover more relation variants (Attached new experiments, Section E.3, Figures 19a-19f). Apart from the augmentations listed above, we also confirmed that CoDeC successfully detects training on samples rephrased using GPT-5 and training on multiple data mixes.
>
> We observe that CoDeC scores remain consistent across identically distributed datasets (e.g., The Pile training/test splits), regardless of which was used for training. Similarly, in our evaluation of OLMo training checkpoints (Figure 4), CoDeC scores converge to their final values after only ~2% of training, indicating reliable detection even though the model saw just a tiny fraction of each dataset.
>
> > The paper mentions thresholding and comparison against other models. Could the authors elaborate on practical guidelines for selecting these thresholds in real-world evaluation scenarios?
>
> This is an important point, as there are multiple ways the CoDeC scores can be utilized. If a binary label is needed, the simplest off-the-shelf approach is to interpret scores above 80% as indicative of contamination, while lower scores suggest insufficient evidence. This threshold is consistent across models and conservative enough to minimize false positives.
>
> For a more principled approach, we recommend evaluating a set of “anchor” models – preferably of similar size and known to be uncontaminated (e.g., research models trained on public corpora) – and assessing whether the suspect model is an outlier within their score distribution. The results reported in Appendix D can serve as a reference for such comparisons.
>
> There are also additional ways to leverage CoDeC scores. Since CoDeC effectively measures the remaining capacity for learning the target data, models with lower CoDeC scores (given similar accuracy) are expected to generalize better, as they likely would have achieved higher accuracy with the same exposure. In practice, this interpretation can be particularly useful for enriching benchmark evaluations and for checkpoint selection during training. For such comparisons no threshold is needed.
>
> Following your comments, we have expanded our guidance on interpreting CoDeC scores (Section 3.5, Appendix A.5) and added a remark on score interpretation in the Conclusions. Thank you for raising this point.
>
> > Do the authors have any theoretical guarantees (e.g., consistency or error bounds) on CoDeC scores? Or atleast outline what such analysis might require?
>
> CoDeC models the tendency of the model to get distracted in modeling a target text, upon seeing other related text from the same benchmark. In the simple case of a purely probabilistic language model that only learns bigrams probabilistically, a theoretical model could factor in the contribution of in-context tokens and scale it with respect to certain assumptions on the token-token frequencies seen during training.
>
> While such simplified models do not capture the complexity of modern transformer architectures where logits arise from large‑scale distributed representations and attention mechanisms, the structure of the CoDeC score naturally points to statistical analysis. For a fixed dataset, each target sample either shows a confidence drop or not, which can be treated as a Bernoulli trial. This framing allows to derive exact confidence intervals for the contamination score based on dataset size or the number of sampled contexts, and could form the basis for principled error bounds or consistency guarantees.
>
> Nevertheless, our observations are primarily supported via extensive experiments on a large set of models and datasets. Confirming those empirical insights via theoretical analysis is a promising direction for future work. Following your feedback, we included that remark in the Future Work section.

---

> ### Author Response · Authors · 2025-11-18
>
> > How sensitive is the method to the number of in-context samples and the randomness of context selection? Would deterministic context selection (e.g., nearest neighbors) further improve stability?
>
> We thank the reviewer for this insightful question. Context selection is indeed central to the effectiveness of CoDeC. In our design, we sample a single context example randomly from the dataset, which provides a representative view of the overall distribution while keeping the method simple. As shown in our ablations (Section 3.4), the variance introduced by random sampling is negligible except for extremely small datasets.
>
> You are right that deterministic strategies such as nearest‑neighbor selection could further influence the trade‑off between (1) the positive effect of providing informative context and (2) the potential disruption of memorization. Our additional rebuttal experiments (Section E.5, Figures 23a-23b) show that nearest‑neighbor contexts increased model confidence, reducing scores for unseen datasets to nearly 0%. However, scores for training datasets also decreased, leaving the gap mostly unchanged.
>
> While integrating of similarity‑based sampling into the CoDeC pipeline requires additional computation for representation search and clustering, it may improve score stability and enhance robustness even against adversarial datasets. We appreciate the Reviewer suggestion and see it as a promising direction for future work.
>
> > Although contamination transfer across related datasets (e.g., MMLU) is explored, adversarial or edge cases (e.g., near-identical but unseen datasets, noisy mixtures) are only lightly discussed.
>
> Following your suggestion, in the attached additional experiments (Section E.1), we analyze in detail adversarial and degenerate datasets. We find that while it is easy to design datasets that consistently produce low scores, creating ones that always produce high scores is significantly harder (which ensures that CoDeC mitigates the risk of false positives). Furthermore, such effects only become pronounced under extreme conditions (e.g., reducing the dataset to four distinct samples repeated many times).
>
> We also expanded our contamination transfer experiments to cover more near-identical but unseen datasets (Section E.3, Figures 19a-19f), including rephrases, various mixtures, noisy datasets, etc. In particular, we observed that models obtain nearly identical scores on identically distributed datasets (such as Pile train/test splits), regardless of which part was seen during training.
>
> We acknowledge that, like any metric, CoDeC can be influenced by adversarial or degenerate datasets. However, they do not hinder the effectiveness of CoDeC in practice, as the target dataset is always the user choice. Just like it is possible to construct benchmarks that yield always low or always high scores regardless of true model capabilities, such datasets simply serve no purpose.
>
> Specifically, to analyze the adversarial datasets we conducted the following experiments:
> - Models get 0% score if we create a dataset of a single example repeated 100 times; both when that example was seen or not; even when we introduce a small noise to each copy.
> - Such an effect is perceptible only when we reduce a real dataset size to as few as 4 samples repeated 32 times each.
> - If we mix together 4 unrelated datasets on which models got <20%, they start getting ~50% on the mix.
> - Evaluating contamination on a purely random dataset always gives 0% scores.
> - Using similarity-based context sampling shows greater robustness to unrelated data mixes.
>
> > The method might conflate contamination with related distribution overlap, as the authors acknowledge, but do not quantitatively evaluate how severe this is.
>
> One of the key features of CoDeC is that it does not test solely the exact training membership, but captures also the contamination from related data (such as rephrasing, random cropping, noise injection, and other augmentations). This way, we may capture data leaks, even augmented, making CoDeC very useful and hard to bypass. Our contamination transfer experiments (Section 3.3, Appendix C.5, and additional rebuttal experiments in Section E.3) quantify this effect on multiple related distributions.
>
> There are multiple data properties that influence contamination. Even training on QA benchmarks slightly increases contamination with other QA benchmarks. We view it as a desired property – from an evaluation perspective, given two models with similar accuracy on a QA benchmark, the one with prior exposure to QA data likely leveraged that experience, whereas the other relied solely on general capabilities. CoDeC helps disentangle such cases, providing insight into the true source of benchmark performance.

---

> ### Author Response · Authors · 2025-11-18
>
> > Some elements overlap conceptually with loss-based or entropy-based detection. Positioning CoDeC more clearly in terms of unique advantages and trade-offs would help.
>
> We thank the reviewer for this suggestion. While CoDeC shares some conceptual elements with loss‑based and entropy‑based detection, it offers several distinctive features:
> - Threshold‑free criterion: Rather than tuning model‑specific thresholds for loss or entropy, CoDeC evaluates whether the context impact is positive or negative – a universal, model‑independent decision rule essential for practical deployment.
> - Self‑calibration: CoDeC regularizes loss using the target model’s own predictions, adapting naturally to its internal confidence distribution.
> - No external dependencies: In contrast to reference‑based or entropy‑based methods requiring auxiliary models or datasets, CoDeC derives baselines from the same model, avoiding reliance on outside resources.
> - Minimal assumptions: Works leveraging dataset structure rely on strong requirements: access to non-contaminated identically distributed data, full knowledge of target mode training, etc. CoDeC makes no such assumptions and can be applied to arbitrary models and data off-the-shelf.
>
> To our best knowledge, the key idea behind CoDeC (using same-model same-dataset predictions as a baseline for loss calibration) is novel. No other approach provides such good separation between seen and unseen data with such a lightweight and simple pipeline. To highlight the advantage of our self-regularized pipeline, we included loss-based and entropy-based methods as baselines in our main validation plot. The advantage of CoDeC is clear.
>
> Following your suggestion, we revised our Related Works section to better highlight the unique advantages of CoDeC. Thank you for bringing this to our attention.
>
> > The paper claims with the help of CMA that LLMs
>
> Could you please clarify this comment, as it appears to be partially missing?

---

> > ### Comment · Reviewer_sLbA · 2025-11-27
> >
> > Thank you for your detailed rebuttal. I appreciate the clarifications you provided in your response.
> >
> > Regarding the incomplete sentence you referenced - "The paper claims with the help of CMA that LLMs…", it was mistakenly left incomplete in my original draft and does not reflect any criticism. You may disregard it completely.
> >
> > I would also like to acknowledge that all of my questions have now been addressed. Based on your clarifications and the strength of the work, I will be increasing my score for the paper to 6.

---

> > > ### Author Response · Authors · 2025-11-27
> > >
> > > Dear Reviewer sLbA,
> > >
> > > Thank you for your response and for increasing your score. We are glad to hear that our rebuttal successfully addressed all your questions. Thank you for your time and constructive feedback.

---

### Official Review · Reviewer_xj5F · 2025-11-01

**Soundness:** 3
**Presentation:** 3
**Contribution:** 3
**Rating:** 6
**Confidence:** 3

**Summary:**

The paper introduces CoDeC, a dataset-level method to detect training-data contamination in LLMs by measuring how few-shot, same-dataset context changes model confidence on a target sample. If added context improves confidence, the dataset is likely unseen; if it lowers confidence, the model likely memorized the dataset or closely related data. The per-sample confidence shift Δ(x) is aggregated into a dataset contamination score SCoDeC(D) = (1/N)∑1[Δ(x)<0], requiring only two forward passes per sample and gray-box access to token probabilities. The authors show near-perfect separation of seen vs. unseen datasets (dataset-level AUC ≈ 99.9%) across many models, and provide analyses on training dynamics, finetuning-induced contamination, and robustness.

**Strengths:**

1) Clear, elegant idea with strong intuition: leverage in-context learning as a probe—unseen datasets benefit from added context, while memorized datasets get perturbed—yielding a direct, interpretable signal.

2) Practicality & efficiency: model- and dataset-agnostic; only two forward passes per sample; no threshold tuning or access to training corpora needed.

3) Simple, interpretable metric: a percentage score (fraction of samples with negative Δ) that aligns with practitioner intuition and enables straightforward ranking across datasets.

4) Strong empirical evidence: near-perfect dataset-level AUC over diverse models; analyses show early emergence of contamination in training, finetuning-induced contamination, and size trends (larger models memorizing less).

5) Scope beyond strict membership inference: also detects contamination via related/shadow distributions, broadening practical utility for benchmark hygiene.

**Weaknesses:**

1) Adversarial/degenerate datasets: repeated or highly heterogeneous mixtures can distort the score; the paper notes such edge cases but they remain a limitation for fully automatic use.

2) Calibration to “absolute” labels: SCoDeC is excellent for ranking datasets by contamination risk, but stakeholders may still desire thresholded decisions (ACC/PR/F1). Adding a recommended thresholding recipe could help some users.

3) Model-family anomalies: certain heavily instruction-optimized models behave atypically under CoDeC (e.g., chat/trace behaviors perturb logits broadly), suggesting architecture/task biases may require special handling.

**Questions:**

1) Could you provide a small, principled thresholding guide (e.g., validated percentiles on held-out datasets) for users who need binary “contaminated/not” decisions in audits?

2) How does SCoDeC behave under various context sizes n across different dataset lengths/formats? A brief cost–benefit curve would aid deployment.

3) For instruction-tuned/chat-optimized models that yield universally high scores, can lightweight decoding settings (e.g., stop tokens, no-think modes) mitigate anomalies?

---

> ### Author Response · Authors · 2025-11-18
>
> Dear Reviewer xj5F,
>
> We thank you for your feedback and are pleased that you consider our method both elegant and practical. Below, we address all of your questions and comments.
>
> > Could you provide a small, principled thresholding guide (e.g., validated percentiles on held-out datasets) for users who need binary “contaminated/not” decisions in audits?
> > Calibration to “absolute” labels: SCoDeC is excellent for ranking datasets by contamination risk, but stakeholders may still desire thresholded decisions (ACC/PR/F1). Adding a recommended thresholding recipe could help some users.
>
> For robust contamination assessment, we suggest combining absolute thresholds with
> reference-based comparisons, as follows:
> - For users requiring binary “contaminated/not” decisions, the simplest approach is to treat scores above 80% as indicative of contamination, and lower scores as insufficient evidence that requires more investigation.
> - For a more principled approach, we recommend evaluating a set of “anchor” models – preferably of similar size and known to be uncontaminated (e.g., research models trained on public corpora, or models known to be released before a benchmark release) – and assessing whether the suspect model is an outlier within their score distribution. The leaderboard reported in Appendix D can serve as a reference for such comparisons.
>
> We agree that our guidance on interpreting scores (Section 3.5, Appendix A.5) could be highlighted more. In response, we have expanded the discussion in the Appendix to provide clearer recommendations  and added a remark on score interpretation in the Conclusions.
>
> > How does SCoDeC behave under various context sizes n across different dataset lengths/formats?
>
> That is an important point. We found that using too long inputs is not necessary, but on the other hand, using too short inputs somewhat increases the variance of scores. Our new experiments (Section E.2, Figures 18a-18d) show that overly long inputs are unnecessary, while very short inputs increase score variance and bias results slightly toward 50%. No performance gains were observed beyond moderate input lengths.
>
> In practice, we cap each sample at 2048 characters and discard any below 100 characters to ensure stable results. Evaluating 1000 samples is sufficient for negligible variance, requiring only 2000 forward passes over inputs of at most 4096 characters – a very low computational cost given that no generation is involved. We therefore do not recommend reducing sample length as a cost‑saving measure.
>
> You are right that for shorter prompts providing additional context reduces variance (as shown in Figure 9). Context selection by total character or token count is particularly promising for datasets with highly variable sample lengths, though in our experiments it was not necessary.
>
> > Model-family anomalies: certain heavily instruction-optimized models behave atypically under CoDeC (e.g., chat/trace behaviors perturb logits broadly), suggesting architecture/task biases may require special handling.
> > For instruction-tuned/chat-optimized models that yield universally high scores, can lightweight decoding settings (e.g., stop tokens, no-think modes) mitigate anomalies?
>
> In our main validation (Figure 3), CoDeC demonstrates consistent separation across a wide range of model types – including base, instruction-tuned, reasoning-focused, transformers, recurrent, and hybrid architectures – indicating that training style or architecture does not generally hinder its effectiveness. In the leaderboard results (Appendix D), instruction-tuned variants achieve scores closely aligned with their corresponding base models, suggesting that instruction tuning itself is not a source of systematic bias in CoDeC measurements.
>
> The only notable exception we observed is GPT‑OSS, which appears to lack standard text completion behavior: even when provided with identical context, its confidence drops unexpectedly. This anomaly is unique among the models we tested and is unrelated to instruction tuning itself. For example, we evaluate models like Qwen3-30B-A3B-Instruct-2507, Phi-4-reasoning-plus, Mistral-Large-Instruct-2411 (among others) and none of them exhibit such anomalies.
>
> We would like to clarify that CoDeC scores are computed directly from target text logits, without any generation step. As such, they are independent of decoding strategies (e.g., stop tokens, “no-think” modes) or other inference-time settings.

---

> ### Author Response · Authors · 2025-11-18
>
> > Adversarial/degenerate datasets: repeated or highly heterogeneous mixtures can distort the score; the paper notes such edge cases but they remain a limitation for fully automatic use.
>
> We acknowledge that, like any metric, CoDeC can be influenced by adversarial or degenerate datasets. However, they do not hinder the effectiveness of CoDeC in practice, as the target dataset is always the user choice. Just like it is possible to construct benchmarks that yield always low or always high scores regardless of true model capabilities, such datasets simply serve no purpose.
>
> In the attached additional experiments, we analyze in detail adversarial and degenerate datasets (Section E.1). We find that while it is easy to design datasets that consistently produce low scores, creating ones that always produce high scores is significantly harder (which ensures that CoDeC mitigates the risk of false positives). Furthermore, such effects only become pronounced under extreme conditions (e.g., reducing the dataset to four distinct samples repeated many times).
>
> Specifically, we conducted the following experiments:
> - Models get 0% score if we create a dataset of a single example repeated 100 times; both when that example was seen or not; even when we introduce a small noise to each copy.
> - Such an effect is perceptible only when we reduce a real dataset size to as few as 4 samples repeated 32 times each.
> - If we mix together 4 unrelated datasets on which models got <20%, they start getting ~50% on the mix.
> - Evaluating contamination on a purely random dataset always gives 0% scores.
> - Using similarity-based context sampling shows greater robustness to unrelated data mixes.

---

> ### Author Response · Authors · 2025-11-30
>
> Dear Reviewer xj5F,
>
> we would like to thank you for your time and commitment to reviewing our paper, and for your constructive feedback, particularly your suggestions to expand the discussion of score interpretation and to add a short, principled thresholding guide, which we believe have made our paper stronger and more useful for practitioners.
>
> Once again, we thank you for the positive reception of our research and for highlighting so many strengths. We hope that our responses have addressed all your remaining concerns and have further convinced you of the high quality of our method.

---

### Author Response · Authors · 2025-11-18

We thank all Reviewers for their detailed analysis and insightful, constructive feedback. We are encouraged that our idea was described as clear and elegant (all Reviewers), supported by extensive experiments (sLbA, 7wL6) providing strong empirical evidence (xj5F), and recognized as practical and scalable (xj5F, sLbA). We also appreciate your comments that the paper is well‑written (sLbA, 7wL6) and important to the field (7wL6).

Guided by your feedback, we have substantially expanded our analysis, experiments, and presentation to address your questions. Key updates include:
- Providing motivating examples and additional empirical analysis to clarify the core idea behind CoDeC.
- Enhancing guidance on interpreting CoDeC scores and highlighting best practices in the Conclusions.
- Extending the leaderboard to cover over 40 models. This leaderboard can also further help the reader contextualize CoDeC scores.
- Expanding the examination of scores through ablation studies involving training on related or augmented data (e.g., rephrases, mixed datasets).
- Adding detailed analysis of adversarial and degenerate datasets, quantifying their impact in real settings, and discussing mitigation strategies.
- Analyzing similarity-based deterministic context selection.
- Studying the effect of sample length on CoDeC scores.
- Updating Related Works to better emphasize unique advantages of CoDeC.

All new experiments, charts, and supplementary analyses are gathered in the Appendix E of the paper and will be subsequently integrated into the contents.

We hope these additions fully address your questions. If any points remain unclear or you would like to explore specific topics further, we would be happy to continue the discussion.

---

### Meta-Review · Area_Chair_8tQ3 · 2026-01-23

**Summary:**

This paper introduces CoDeC, a simple and efficient method to detect training data contamination in LLMs by measuring how in-context examples affect model confidence. Reviewers raised concerns about vulnerability to adversarial or degenerate datasets (e.g., extreme repetition causing false negatives), atypical logit behavior in certain instruction-tuned/chat models, lack of strong theoretical guarantees (relying on empirical intuition), heuristic thresholding limiting high-stakes binary use, insufficient robustness to rephrased/stylistically similar unseen data, unclear differentiation from perplexity/loss baselines, and need for broader edge-case coverage (partial contamination, mixed training). In the rebuttal, the authors added extensive new experiments—including rephrasing tests, adversarial/mitigation cases, mixed-training results, ablations on context size/sampling, contamination transfer analysis, anchor-model comparisons, and practical thresholding guidance (e.g., >80% as contamination flag)—along with clarifications on method motivation, visualizations, and positioning. The reviewers were actively engaged, with one increasing their score from 4 to 6, resulting in uniform 6s (marginally above threshold, acceptable either way). By reviewing the authors’ responses and reviewers’ feedback, I believe most concerns have been effectively addressed through strengthened empirical results, supporting a positive rate.

**Reviewer Concerns:**

The rebuttal convincingly addressed the concerns of adversarial robustness, rephrasing/mixed training coverage, thresholding/practical use, model anomalies (shown to be non-systematic, mostly isolated), and differentiation/empirical grounding.

Still outstanding concerns
Absence of formal theoretical guarantees or error bounds (e.g., probabilistic analysis of false positives/negatives, consistency proofs) — rebuttal framed scores as Bernoulli trials for confidence intervals and noted future theoretical work, but provided no new proofs or bounds; reviewers viewed this as common in empirical ICL papers but still a gap.
Fully principled/validated thresholding for audit scenarios (e.g., validated percentiles on held-out datasets, ACC/PR/F1 metrics for binary decisions) — rebuttal gave heuristic 80% + anchor guidance, but no held-out validation or quantitative metrics for threshold performance, leaving high-stakes audit use somewhat heuristic.
Complete resolution of adversarial limitations for fully automatic/trustworthy deployment — rebuttal mitigated via analysis (harder to fake high cleanliness, extremes predictable), but acknowledged degenerate cases remain a practical limitation (cannot fully eliminate distortion risk in automated settings without user checks).

Overall, the rebuttal resolved the large majority of concerns through substantial new empirical evidence and transparency, leading to improved reviewer consensus. The few remaining issues are mostly theoretical depth and absolute thresholding rigor, which are positioned as future directions rather than critical blockers.

**Reviewer Scores:**

Reviewer xj5F (initial/final: 6)： Main concerns (adversarial cases, thresholding, instruction-tuned anomalies, context sensitivity) were directly addressed.

Reviewer sLbA (initial: 4 → final: 6)
All questions were explicitly resolved in the rebuttal. The reviewer will increase the score.

Reviewer 7wL6  (initial/final: 6)
Specific questions (rephrasing, mixed training, capability correlation) were fully answered with targeted experiments.

---

### Decision · Program_Chairs · 2026-01-26

Accept (Poster)